# Basal Diet Determined Long-Term Composition of the Gut Microbiome and Mouse Phenotype to a Greater Extent than Fecal Microbiome Transfer from Lean or Obese Human Donors

**DOI:** 10.3390/nu11071630

**Published:** 2019-07-17

**Authors:** Daphne M. Rodriguez, Abby D. Benninghoff, Niklas D.J. Aardema, Sumira Phatak, Korry J. Hintze

**Affiliations:** 1Department of Animal, Dairy and Veterinary Sciences, Utah State University, Logan, UT 84322, USA; 2USTAR Applied Nutrition Research, Utah State University, Logan, UT 84322, USA; 3Department of Nutrition, Dietetics and Food Sciences, Utah State University, Logan, UT 84322, USA

**Keywords:** obesity, gut microbiome, western diet, fecal microbiome transfer

## Abstract

The Western dietary pattern can alter the gut microbiome and cause obesity and metabolic disorders. To examine the interactions between diet, the microbiome, and obesity, we transplanted gut microbiota from lean or obese human donors into mice fed one of three diets for 22 weeks: (1) a control AIN93G diet; (2) the total Western diet (TWD), which mimics the American diet; or (3) a 45% high-fat diet-induced obesity (DIO) diet. We hypothesized that a fecal microbiome transfer (FMT) from obese donors would lead to an obese phenotype and aberrant glucose metabolism in recipient mice that would be exacerbated by consumption of the TWD or DIO diets. Prior to the FMT, the native microbiome was depleted using an established broad-spectrum antibiotic protocol. Interestingly, the human donor body type microbiome did not significantly affect final body weight or body composition in mice fed any of the experimental diets. Beta diversity analysis and linear discriminant analysis with effect size (LEfSe) showed that mice that received an FMT from obese donors had a significantly different microbiome compared to mice that received an FMT from lean donors. However, after 22 weeks, diet influenced the microbiome composition irrespective of donor body type, suggesting that diet is a key variable in the shaping of the gut microbiome after FMT.

## 1. Introduction

Obesity rates in humans have increased remarkably in the past several decades, making this disease arguably the greatest current health challenge facing Western societies. According to 2018 data from the Centers for Disease Control and Prevention, approximately 36% of adults and 16% of children and adolescents in the United States are considered obese, which is defined as having a body mass index (BMI) greater than 30. The etiology of obesity is multifactorial, as the disease is typically caused by a combination of genetic and environmental factors, including dietary patterns and a sedentary lifestyle. Obesity is linked to a variety of medical problems, including hypertension, coronary artery disease, cancer, and type 2 diabetes. Individuals that are overweight also have a higher risk of developing esophageal, colorectal, or gallbladder cancers, among others [1]. Obesity and its related pathologies can be grouped broadly as metabolic syndrome and consists of four general characteristics, including central obesity, high triglycerides, high blood pressure, and high fasting plasma glucose [2]. As a function of increasing obesity rates, the prevalence of metabolic syndrome has also increased, and it is estimated that 34% of the United States population fits this classification [3]. Because of its association with chronic disease, the mitigation of obesity has been the subject of many scientific inquiries.

The gut microbiome is the most diverse and complex community of microorganisms in the body, consisting of over one-thousand bacteria species [4]. The relationships between these bacteria and the host are generally commensal or symbiotic in nature. Humans derive approximately 10% of their daily energy intake from microbial fermentation of indigestible food components to generate short-chain fatty acids (SCFAs) [5]. Substantial changes in the composition of the microbial community, or dysbiosis, can lead to adverse health effects for the host. A shift in microbiota composition can also be associated with disease, including metabolic syndrome, inflammatory bowel syndrome, and colorectal cancer [6,7,8]. Lower microbial gene diversity has been reported in obese individuals compared to their lean counterparts [9,10]. A shift in the ratio toward Bacteroidetes when energy is restricted has been equivocal [11,12]. However, recent meta-analyses of human data have cast doubt on the concept of an “obese” microbiome [13,14]. Meta-analyses of 10 studies using random forest machine learning models reported a 33% to 65% median accuracy for the predictability of obesity based on microbiome composition [14]. In addition, Finucane et al. observed no association between BMI and gut microbiota composition and diversity [13]. Taken together, these data suggest that weight loss through energy restriction may change the microbiota composition, but weight-stable obese humans may not have a characteristic microbiome compared to lean counterparts.

The interaction between the microbiome and obesity has been the subject of numerous preclinical investigations employing a fecal microbiome transfer (FMT) approach. Germ-free mice inoculated with microbiota from obese mice had increased cecal concentrations of butyrate and acetate and decreased fecal energy content, thought to be driven by increased *Firmicutes* [15]. When mice inoculated with obese microbiota were cohoused with mice harboring lean microbiota, they were protected from increased weight gain and development of an obesity-associated metabolic phenotype [16]. Conversely, obese mice fed a high-fat diet supplemented with microbiota from lean mice had altered gut microbiota diversity as well as increased body weight compared to counterparts [17]. Rabot et al. observed that mice transplanted with microbiota from mice that were either affected or resistant to diet-induced obesity and fed a high-fat diet gained weight independently from the microbiota donor phenotype [18]. The limited data concerning obesity and FMT in humans have suggested that the microbiome may play a more limited role in weight gain than is suggested by some preclinical data. In two human studies, insulin resistance was improved in individuals who received a fecal transplant from healthy, lean donors [19,20]. However, in both studies, obese subjects receiving microbiota from lean donors did not lose any weight, suggesting that the change in the insulin resistance phenotype was related to changes in the microbiome and not weight loss.

Diet is a key factor in obesity and can also shape the composition of the gut microbiome. For example, different cultures that consume distinct diets have been reported to have different microbiomes [21,22]. The Western style dietary pattern is characterized as energy-rich and nutrient-poor with a high consumption of red meat, animal fat, and sugar coupled with low fiber intake [23]. Chronic deficiencies of essential micronutrients typified by the Western dietary pattern can lead to chronic diseases [24,25]. Diets high in sugar and fat can disrupt metabolism and homeostasis, causing microbiome dysbiosis along with obesity and metabolic syndrome [26]. Furthermore, changes in diet can rapidly cause changes in composition of the gut microbiome [27]. Collectively, these results demonstrate that diets might have a greater influence on gut microbiota composition compared to other factors [18]. Thus, the basal diet administered to experimental animals must be an important consideration for preclinical microbiome studies. Typically, the AIN93G purified diet is used as a standardized diet to promote animal health, whereas diet-induced obesity (DIO) formulations are used to model the Western dietary pattern [28]. However, most of these simple high-fat diets do not emulate the macro- and micronutrient profile typical of the Western dietary pattern, and thus they are generally not relevant for at-risk human populations. To address this issue, Hintze et al. developed the total Western diet (TWD) for rodents by translating the 50th percentile of micro- and macronutrient intakes reported in the National Health and Nutrition Examination Survey (NHANES) using an energy density approach [29,30]. 

Although the interaction between the gut microbiome and obesity has been well studied, little is known about the effects of different basal diet patterns on the composition of the gut microbiome, specifically in studies employing FMT from potential donors with different disease phenotypes. The primary objective of this study was to determine whether FMT from lean or obese human donors would confer these host traits to mice fed either a standard diet, a high-fat diet, or a Western-type diet for 22 weeks. This study design emulated FMT from lean or obese humans consuming different diets as opposed to previous studies with germ-free mice fed for shorter periods time. We hypothesized that mice receiving fecal bacteria from obese humans would develop an obese phenotype with symptoms of metabolic syndrome that would be enhanced by consumption of either the TWD or a 45% fat diet-induced obesity (DIO) diet.

## 2. Materials and Methods

### 2.1. Human Fecal Microbiota Collection

The Utah State University Institutional Review Board approved procedures for the collection of human stool samples from lean or obese donors (protocol #7454). Donors were asked to complete an online survey to indicate their interest in participation. Donors were then contacted to complete an interview at the Center for Human Nutrition Studies clinic office. During these interviews, participants completed a health history questionnaire and were educated regarding informed consent, and the subjects’ age, weight, height, and waist circumference measurements were recorded. Exclusion criteria included age <18 years, antibiotic use within the past three months, or a recent diagnosis of diabetes. Participants collected a fecal sample in a provided plastic, opaque container, which was stored frozen until delivery to the clinic office. Donors who provided fecal samples were compensated $20 for participation. Upon receipt at the clinic, fecal material was stored at −80 °C until use in the animal study. Donors were categorized according to their BMI, which was calculated as weight in kilograms divided by height in meters, and a BMI <25 was defined as lean and >30 as obese. Waist circumference was also used as a criterion for inclusion, with values <90 cm considered lean and values >100 cm considered obese. The waist circumference metric is useful to avoid the categorization of individuals with a high BMI attributed to high lean muscle mass [31]. Three human donors were assigned to each category, and their samples were then blinded with an alias ID to maintain confidentiality with research personnel. Donor parameters are summarized in Appendix A. 

### 2.2. Experimental Animals

The Utah State University Institutional Animal Care and Use Committee approved all procedures for the handling and treatment of mice used in this study (protocol #2491). Animals were housed in the Laboratory Animal Research Center at Utah State University, which is accredited by the Association for Assessment and Accreditation of Laboratory Animal Care. Mice were maintained in a pathogen-free vivarium at 18 to 23 °C with a 12:12 h dark/light cycle and humidity maintained between 20% and 50%. Mice were housed individually in sterile microisolator cages with Bed-o’Cobs® 1/4 bedding (Andersons, Cincinnati, OH, USA), supplied with HEPA-filtered air, and provided with autoclaved water. Cages were autoclaved weekly. Eight-week-old male C57/BL6 mice were obtained from Jackson Laboratories (Bar Harbor, ME, USA) and were quarantined for a week for acclimation, during which time they were provided free access to an AIN93G diet and plain drinking water. Due to the limited size of the animal study and the reasonable expectation that male and female mice may respond differently to obesogenic- or metabolic syndrome-inducing diets [32], only male recipient mice were included in the experiment design.

### 2.3. Experimental Diets

Experimental diets were obtained from Envigo (Chicago, IL, USA). Test diets were irradiated and pelleted at Envigo and then stored at 4 °C prior to use. Three test diets were used to determine the impact of a basal diet on the microbiome and phenotypic parameters following FMT from human donors, including: (1) The AIN93G diet (AIN, Cat. No. TD.94045), which was formulated to support proper growth and reproduction in rodents; (2) a commercial diet-induced obesity (DIO) diet (Cat. No. TD.06415) that contained 45% of energy as lard; and (3) the total Western diet (TWD) (Cat. No. TD.110424), which models typical U.S. nutrient intakes on an energy density basis, as described previously [29]. Diet formulations are provided in Appendix A.

### 2.4. Microbiota Depletion and Fecal Microbiota Transfer from Human Donors

Previously, our group developed a broad-scope antibiotic method for depleting the mouse microbiome followed by FMT from human donors to establish a microbiome closely related to the donor [30]. Using this method, we showed that recipient mice had a microbiome more closely related to the donor with respect to species composition and metabolic activity. Thus, this same approach was employed in the present study. Briefly, mice were orally gavaged every 12 h with 1 mg/kg amphotericin-B for 3 days to prevent fungal infections. For the next 14 days, a broad-spectrum antibiotic cocktail containing 50 mg/kg vancomycin, 100 mg/kg neomycin, 100 mg/kg metronidazole, and 1 mg/kg amphotericin-B was administered by oral gavage every 12 h. During this period, mice were provided water containing 1g/L of ampicillin ad libitum. This antibiotic protocol has been shown previously to deplete the mouse gut microbiome [33]. Next, 12 h after the last antibiotic treatment, each mouse was dosed by oral gavage with fecal matter diluted in sterile saline (1 g/mL) from its assigned human donor. FMT was initiated 12 h after the last antibiotic treatment to limit the potential growth of any residual resident bacteria in the recipient mice. FMT occurred once per week for the next 4 weeks according to the protocol of Hintze et al. [30].

### 2.5. Experimental Design and Phenotype Assessment

Mice were randomly assigned to one of the six human (3 lean and 3 obese males) donors and one of the three diets described above (Figure 1). A total of 144 mice were used in the study, with an initial 48 mice allocated for each experimental diet. After completion of the antibiotic protocol, during which some mice died due to the stress of repeated oral gavage, the remaining numbers of mice per diet group were *n* = 43, AIN diet; *n* = 42, DIO diet; and *n* = 41, TWD. Following the antibiotic treatment, six to nine mice were assigned to each lean or obese donor subgroup. Fecal samples were collected weekly and stored at −80 °C. Because the microbiome composition may change depending on the time of the stool collection [34,35,36], care was taken to collect feces at the same time of day for each time point. Fresh food was provided weekly, and food consumption was monitored by differential weight. Total energy intake was calculated based on the sum of weekly estimated food intake using energy density values of 3.8, 4.6, or 4.4 kcal/kg diet for the AIN, DIO, and TWD diets, respectively. Mouse body weight was also measured weekly. At week 20, fasting glucose level and response to an oral glucose tolerance test (oGTT) was obtained as previously described [37]. Briefly, following a fasting period of 6 h, approximately 2 μL of blood was drawn from a 1-mm cut on the tail tip. Glucose was measured using a standard glucose meter (Total Diabetes Supply, Boca Raton, FL, USA) in triplicate for each animal. Fasting glucose was determined immediately following the fasting period. Then, mice were administered a bolus of 10 mg/kg glucose by oral gavage, and blood glucose levels were measured in triplicate for samples obtained via tail tip puncture at 15, 30, 45, 60, 90, and 120 min postgavage. The positive incremental area under the curve (AUC) for oral glucose tolerance was calculated by subtracting time 0 glucose concentration (fasting glucose) from all subsequent measurements for each individual mouse. At week 21, body composition (lean mass and fat mass) was measured by magnetic resonance image (MRI) scan (EchoMRI-700; EchoMRI, Houston, TX, USA). 

At week 22, mice were randomized for order of necropsy, which was performed by rapid CO_2_ asphyxiation and cardiac puncture. The liver, cecum, subcutaneous fat pad, and gonadal fat pad were excised, weighed, quick-frozen, and stored at −80 °C. Blood serum was collected by centrifugation (10,000× *g* for 5 min) using serum separation spin tubes (Sarstedt, Newton, NC, USA), aliquoted into triplicate samples, and then stored at −80 °C. 

### 2.6. Microbiota Profiling by 16S rRNA Sequencing

Fecal samples were collected initially from all mice in the study prior to antibiotic treatment. Then, mouse fecal samples were collected from each cage weekly throughout the study or at necropsy directly from the colon and stored at −80 °C. For this study, only the initial fecal samples, the posthuman FMT samples, and the terminal samples were used for sequencing (Figure 1), in addition to samples from each human donor. The QIAamp DNA Stool Mini Kit (Qiagen, Frederick, MD, USA) was used to isolate DNA from mouse fecal pellets and the frozen human donor samples following the manufacturer’s protocol, with one modification. Zirconia/Silica Beads (Fisher Scientific, Waltham, MA, USA) and buffer were added to the sample, which was mechanically disrupted using a homogenizing bead mill (Bead mill 4, Fisher Scientific) for 5 min. Samples were then processed per the protocol with no other deviations. DNA concentration and sample purity were determined by UV spectrophotometry (NanoDrop 2000, Thermo Fisher Scientific, Waltham, MA, USA). All DNA samples were then diluted to 20 ng/ml in tris-EDTA buffer (TE, pH 8.0).

Isolated fecal DNA was amplified using the Roche High Fidelity dNTP Pack according to the manufacturer’s protocol (Millipore Sigma, St. Louis, MO, USA). Each sample was assigned a barcoded primer, which is outlined in Appendix A, and a universal reverse primer. Barcoded primers were directed against the V3 region of the 16S rRNA [38]. PCR amplification was performed using the following protocol: 5 min at 95 °C; 35 cycles of 94 °C for 30 sec, 55 °C for 30 sec, and 72 °C for 90 sec; final annealing at 72 °C for 10 min; and holding at 4°C. Electrophoresis was then performed with the PCR amplicons to confirm a product size of approximately 280 bp. PCR products were subsequently purified using Agencourt AMPure beads (Beckman Coulter, Indianapolis, IN). PCR products were washed with ethanol to eliminate excess primers, nucleotides, and enzymes present in the PCR mix. DNA was eluted from the beads with TE buffer, and DNA concentrations were reconfirmed by spectrophotometry (Microplate Fluorometer 9300-002, Turner BioSystems, Sunnyvale, CA, USA) using the Quant-IT Picogreen dsDNA assay (Thermo Fisher Scientific). Samples were then diluted to 1 ng/μl per sample and pooled by combining PCR products generated with primers 1 through 60 into one tube. Samples were stored at −20 °C until sequencing at the USU Center for Integrated Biosystems Sequencing Core. The sizing of sample pools was verified using the Agilent Tape station and DNA High-Sensitivity tapes and reagents. Pools were then quantified with the Qubit and the High-Sensitivity dsDNA reagents. Samples were sequenced using an Ion Personal Genome Machine (PGM) sequencer with a 318 Chip kit and an Ion PGM Hi-Q View OT2 kit for library preparation (Thermo Fisher Scientific). 

Microbiota sequences were processed through the most current version of QIIME [39]. After quality filtering and sample assignment, sequences were clustered into operational taxonomic units (OTUs) [40] at a 97% sequence similarity against a reference GreenGenes OTU database (gg_13_8_otus) using the open-reference OTU picking approach with UCLUST using pick_open_ref_otus.py workflow script [41]. The most abundant sequence from each cluster was selected as the representative sequence. Chimera artifacts were identified using uchime61 [42] and were excluded from sequence data. Taxonomy and alpha and beta diversity analyses were performed using core_diversity_analyses.py script. Raw operational taxonomical units (OTUs) were normalized to the total number of reads for each sample and then merged by the highest level resolution (to species, if available), family, and phylum taxonomy levels. Alpha and beta diversity were determined for the overall main effect of body type and experimental diet and for the effect of donor body type within each diet group. Alpha diversity measures included number of OTUs (total number of OTUs sequenced), Chao1 richness (number of species represented), Faith’s phylogenetic diversity (phylogenetic distance of species present), and the Shannon index (weighted abundance of species present). Statistical analysis of alpha diversity data is described below. Beta diversity was determined using unweighted (qualitative measure that is sensitive to low abundance features) and weighted (accounts or abundance of species) unifrac distance measures and is represented as principal coordinate plots (PCoA) of the first two coordinates. Beta diversity values among test groups were analyzed by the nonparametric permanova test in QIIME, which partitions a distance matrix among sources of variation in order to describe the strength and significance that a categorical variable has in determining variation in distances. A permanova *p*-value <0.01 for this test was considered statistically significant. 

### 2.7. Statistical Analysis

Statistical analyses for the dependent variables food intake, energy intake, final body weight, organ weights (mass and with respect to body weight), body composition (fat mass, lean mass, fat mass percentage, lean mass percentage), fasting glucose, oGTT area under the curve, and alpha diversity (OTUs, Chao1 richness, Faith’s phylogenetic diversity, Shannon index) were performed using a mixed model with a standard least squares personality and the restricted maximum likelihood (REML) method for random effects (α = 0.05) (JMP v12, SAS Institute Inc., Cary, NC, USA). The robust outlier test (ROUT) was used to identify outliers in these endpoint data, with a conservative *Q* = 1% (GraphPad Prism v7, GraphPad Software, La Jolla, CA, USA) prior to statistical analyses. A log_10_ transformation was applied to datasets that did not fit normal distributions or variance assumptions and to improve data visualization. The mixed model included three main factors (levels): Basal diet (AIN, DIO, or TWD), body type (lean or obese), and human donor ID (L1, L2, L3, O4, O5, O6) nested within the body type (Figure 1). Mouse ID was included in the model as a random factor nested within both human donor ID and body type. As determined a priori, the main effects of diet, body type, and the interactions of these main factors are reported herein. Because human donors were carefully selected from a very small available sample set (6 males selected from a population of 8), this factor could not be defined as random in the statistical model. Thus, main effects for human donor ID are also reported, though interactions between individual donors and diet are not shown, as these interactions were not germane to the original hypothesis. Tukey post hoc tests for multiple comparisons were performed to determine the effect of diet on measured parameters when diet was a significant main factor (noted within text as “diet post hoc” *p*-values). In addition, post hoc comparisons were made to determine the effect of FMT from either lean or obese human donors on measured parameters within each diet group (noted within text as “FMT post hoc” *p*-values). For all of these mixed model analyses and post hoc tests, a significant effect was inferred when *p* < 0.05. The rate of weight gain for mice in experimental groups over the 22-week study was determined by linear regression analysis with analysis of covariance (ANCOVA) to determine whether the rates differed by diet or FMT body type (GraphPad Prism).

Microbiome taxonomic abundance data were analyzed using linear discriminant analysis with effect size (LEfSe), which identifies discriminating bacteria taxa based on both statistical significance and biological relevance [43]. First, the nonparametric Kruskal–Wallis sum-rank test (α = 0.05) identified features with significant differential abundance, followed by linear discriminant analysis to estimate the effect size of each differentially abundant feature. For LEfSe analyses, because the statistical model used (nonparametric) cannot account for nested or random effects, all microbiota data were averaged by human donor. Because LEfSe works optimally in paired comparisons, a series of LEfSe tests was performed to identify discriminating taxa for the main factors diet (AIN vs. DIO, AIN vs. TWD, DIO vs. TWD) or body type (lean vs. obese). Lastly, to mimic the experimental tests performed above for other parameters, paired LEfSe tests were then performed for lean versus obese within each diet group. A significant difference was inferred when *p* < 0.05 with a logarithmic linear discriminant analysis (LDA) score threshold of 2.

ClustVis [44] was used to perform unsupervised, bidirectional hierarchical cluster analyses (HCCs) and principal components analyses (PCAs) using relative abundance data for taxonomic classifications at the family level, including families comprising at least 0.1% of the fecal microbiome. To examine overall patterns in the gut microbiota profiles, the relative abundance data for the lowest annotated taxa in mice averaged by the donor ID as well as the relative abundance data for human donors were compared using a Pearson correlation in R (www.R-project.org). Finally, correlation analyses were also performed to compare the relative abundance of fecal bacteria families (minimum abundance >0.5%) in each recipient mouse to the corresponding phenotype measurements, including final body weight, fat mass, and lean mass (gram and percent of body weight), fasting glucose, and glucose tolerance. 

## 3. Results

### 3.1. Food and Energy Intake

Food intake for mice provided either the AIN or DIO diets was not significantly different, nor was a significant effect of donor body type observed for mice within each diet group (Appendix A). However, over the 20-week feeding period, mice fed the TWD and inoculated with microbiota from lean donors ate 3.1% less food than mice fed the TWD and inoculated with microbiota from obese donors (FMT post hoc *p* = 0.0031) (Appendix A). Energy intake generally reflected the different energy contents of the diet, with mice provided the DIO diet consuming significantly more food energy than their AIN-fed counterparts (*p* < 0.0001), with no apparent differences for either AIN-fed (FMT post hoc *p* = 0.68) or DIO-fed (FMT post hoc *p* = 0.97) mice with respect to FMT from either lean or obese donors. Energy intake in TWD-fed mice reflected the slight difference in food intake, with obese-transferred mice consuming significantly more food energy than their lean-transferred counterparts (FMT post hoc *p* = 0.0031).

### 3.2. Body Weight Gain

A decrease in body weight gain was observed across all experimental groups during the first two weeks of the study, which was likely caused by the antibiotic treatment that all mice received (Figure 2A). By week 3, mice had apparently recovered and began gaining weight again, albeit at apparently different rates, as mice fed the DIO diets gained weight at a rate of 0.79 g/week (diet post hoc *p* < 0.0001) compared to 0.37 or 0.46 g/week for those fed the AIN diet or the TWD, respectively. No significant differences in the rate of weight gain as a consequence of FMT from obese or lean human donors was noted for the AIN (FMT post hoc *p* = 0.96), DIO (FMT post hoc *p* = 0.83), or TWD diets (FMT post hoc *p* = 0.22). At 22 weeks, a significant main effect of diet (*p* < 0.0001) was observed for final body weight, with the mice fed the DIO diet weighing significantly more, 19.6% or 20.8%, than their counterparts fed either the AIN diet or the TWD, respectively (Figure 2B). Body type associated with FMT did not significantly affect final body weight for mice in any of the diet groups. This observation was interesting, given that mice fed the TWD and with bacteria transferred from obese human donors had a higher energy intake compared to their lean counterparts, a difference that was not reflected in the final body weight of those mice. 

### 3.3. Body Composition and Fat Distribution 

A significant main effect of experimental diet was observed for body composition, analyzed either as mass values or when normalized to body weight (Figure 3A–D). However, an FMT from either lean or obese human donors did not affect lean or fat mass in recipient mice. Lean mass was elevated by 2.8% in mice fed the DIO diet compared to those given the AIN diet (diet post hoc *p* = 0.0294), but only on a mass basis (Figure 3A). When considered as a fraction of body weight, lean mass was significantly lower in DIO-fed mice compared to their AIN- and TWD-fed counterparts (diet post hoc *p* < 0.0001) (Figure 3C), reflecting the large increase in fat mass in these animals (Figure 3D). As expected, consumption of the DIO diet markedly increased percent fat by 105% or 60% compared to the AIN or TWD diets, respectively (diet post hoc *p* < 0.0001). In addition, consumption of the TWD increased percent fat mass by 29% compared to the AIN controls (diet post hoc *p* < 0.0001) (Figure 3D). A similar pattern was observed for percentage gonadal fat and subcutaneous fat, which were markedly higher in DIO-fed mice compared to their AIN- and TWD-fed counterparts (diet post hoc *p* < 0.0001) (Figure 3E–F). As a percent of body weight, the sizes of the gonadal and subcutaneous fat deposits also increased by 37% (diet post hoc *p* = 0.0023) and 31% (diet post hoc *p* = 0.0147), respectively, in mice fed the TWD compared to the AIN controls.

### 3.4. Liver and Cecum Weight

Liver weight was not significantly affected by experimental diet or human donor body type (Appendix A). Cecum weight was significantly altered by the human donor body type (*p* = 0.0439), with the cecum weight in mice fed the DIO diet that received FMT from obese human donors about 16% greater than in mice that received bacteria from lean donors (FMT post hoc *p* = 0.0295) (Appendix A). 

### 3.5. Fasting Gglucose and Glucose Tolerance 

Fasting glucose concentrations were not significantly affected by either the experimental diet or the transfer of bacteria from lean or obese human donors (Figure 4). Glucose tolerance, calculated as the area under the curve for the glucose response curves shown in Figure 4, was significantly changed by the experimental diets. Mice fed either the DIO diet (diet post hoc *p* < 0.0001) or the TWD diet (diet post hoc *p* = 0.0020) had significantly impaired glucose tolerance, as reflected in the higher AUC values, compared to mice provided the AIN diet (Figure 4E). However, the transfer of bacteria from lean or obese human donors did not significantly affect glucose tolerance in mice fed any of the test diets.

### 3.6. Taxonomic Composition of Human Donor, Initial Mouse, and Recipient Mouse Fecal Microbiomes

Normalized OTU tables constructed from QIIME biom files for all samples are provided in an online data repository [45]. After quality, chimera, and abundance filtering, 1.8 × 10^7^ sequences were assigned to OTUs using the pick_open_ref_otus command for an average of 46,810 sequences per sample assigned to 2269 OTUs. Of these OTUs, 121 were represented in more than 0.01% of the reads. The sequencing depth for diversity analyses was set to 15,000 sequences. An initial fecal microbiome profile was obtained for all mice in the study before they were dosed with antibiotics or provided with experimental diets (Appendix A). This initial microbiome was dominated by bacteria belonging to either the Bacteroidetes phylum (primarily family S24-7), which comprised 42% of the fecal microbiome, or the Firmicutes phylum (primarily families Lactobacillaceae, Lachnospiraceae, Ruminococcaceae, and unclassified members of the Clostriales order), which accounted for 55% of the sequence reads (Figure 5). The microbiome profiles obtained from lean human donors were similar to that of the initial mouse profile with respect to phyla representation, with Bacteroidetes and Firmicutes accounting for 43%–44% or 52%–55% of sequences, respectively (Figure 5). However, notable differences were evident at higher levels of classification when comparing the lean donor profiles either to the original mouse microbiome or when comparing among the lean human donors. For example, bacteria belonging to the Prevotellaceae family were not detected in the initial mouse microbiome. However, Prevotellaceae (genus: *Prevotella*) was identified in lean human donors, although with high variation among the three subjects, ranging from 6.5% to 35% of the mapped reads (Figure 5). In addition, human lean donor microbiomes included varying relative abundances of Bacteroidaceae (genus: *Bacteroides*), ranging from 5% to 34%. Within the Firmicutes phylum, the most abundant bacteria families in the lean donors included unclassified members of the Clostridiales order (8.6% to 15%) and families Lachnospiraceae (21% to 28%) and Ruminococcaceae (6.9% to 22%). Microbiome profiles for obese human donors were substantially more variable, even at the phylum level (Figure 5A), with donor O5 having a markedly different profile (49% Bacteroidetes and 40% Firmicutes) that was more similar to the lean donors compared to donors O4 and O6, which both had substantially reduced relative abundances of Bacteroidetes (19% and 22%, respectively) and higher abundances of Firmicutes (78% and 75%, respectively). As with the lean donors, each profile was notably more distinct when examining taxonomic classifications at higher levels. Within the Bacteroidetes phylum, *Prevotella* was detected in only two of the obese donors (O5 and O6), and the relative abundance of the Bacteroidaceae was also highly variable, ranging from 11% to 29% (Figure 5). Within the Firmicutes phylum, abundances of Ruminococcaceae were more consistent (13% to 20%), while unclassified bacteria for the Clostridia order and members of the Lachnospiraceae family were substantially more abundant in donors O4 and O6 (21% to 26% and 35% to 33%, respectively) compared to donor O5 (7.1% and 8.9%, respectively).

After four weekly FMTs from lean or obese human donors, the fecal microbiome profiles of recipient mice were examined (Figure 5, Appendix A). An inspection of the taxonomic profile for these FMT recipient mice showed that their microbiomes were markedly different from any of the human donors, regardless of donor body type or the initial mouse microbiome. At the post-FMT time point, we identified more discriminating taxa related to the body type of the human donors than were associated exclusively with the experimental diet. Of note was the apparent increase in relative abundance of the Verrucomicrobia phylum (species: *Akkermansia muciniphila*) in all groups after fecal transfer (17% to 23% in lean-FMT recipient mice and 7.3% to 13% in obese-FMT recipients) compared to both human donors (0% to 1.7%) and the initial mouse microbiome (0.2%) (Figure 5, Appendix A). Results of LEfSe analyses pointed to *Akkermansia* as discriminating recipients of lean donor microbiomes when considering all samples irrespective of basal diet (Appendix A), although *A. muciniphila* was identified as discriminating for lean recipients only within the AIN diet group (Figure 6A).

Notable differences in the population of bacteria in the Firmicutes phylum were also observed in mice at the post-FMT time point. For example, the Erysipelotrichaceae family was more abundant in recipient mice post-FMT (5.4% to 6.6%) compared to human donors (0.2% to 1.8%) and the mouse initial microbiome (0.6%) (Figure 5, Appendix A). Within this family, genus *Holdemania* was identified as a discriminating taxon for obese-FMT recipient mice fed each of the experimental diets (Figure 6). Within the Lachnospiraceae family, genus *Coprococcus* was identified by LEfSe analysis as a discriminating taxon for obese-FMT recipient mice fed each of the three test diets, while genus *Ruminococcus* discriminated obese-FMT recipient mice from their lean counterparts fed the AIN and TWD diets only (Figure 6). 

Other key differences associated with diet or body type were noted for bacteria belonging to other phyla. Of note, while the genus *Prevotella* comprised a substantial portion of the Bacteroidetes phylum in most of the human donors, *Prevotella* was not detected in any of the recipient mice following FMT (Figure 5, Appendix A). Within the Actinobacteria phylum, *Collinsella aerofaciens* discriminated lean-FMT mice from their obese counterparts fed either an AIN or DIO diet (Figure 6A–B). Changes in members of the Bacteroidetes phylum were also observed at the post-FMT time point, including for the genus *Bacteroides*, which was associated with obese-FMT mice fed the DIO diet (Figure 6B). Alternatively, LEfSe analysis identified Barnesiellaceae as discriminating lean from obese recipients in mice fed the TWD diet only (Figure 6C). In addition, within the Proteobacteria phylum, bacteria in the order Enterobacteriaceae were identified as discriminating lean-FMT recipients from obese-FMT recipients fed the AIN diet (Figure 6A).

At the post-FMT time point, fewer notable differences were evident for the experimental diet exclusive of contribution of the human donor body type. Within the Bacteroidetes phylum, the order Bacteroidales (most abundant families included Bacteroidaceae, Barnesiellaceae, Odoribacteraceae, Rikenellacaceae) was identified as discriminating between mice fed the AIN or TWD diets compared to those fed the DIO diet (Appendix A), though few specific differences for higher level classifications within Bacteroidales were noted among the different diet groups. Within the Firmicutes phylum, Enterococcaceae abundance was notably greater among all groups (0.66% to 2.5%) in recipient mice post-FMT compared to human donors (all <0.01%) and the original mouse microbiome (0.15%) (Figure 5). Other Firmicutes that discriminated the DIO-fed mice from those given an AIN diet included Streptococcaceae (genus: *Lactococcus*). No discriminating features were identified when comparing the microbiomes of AIN-fed mice to those fed the TWD diet.

At the end of the 22-week study, the fecal microbiomes of all mice were again examined (Figure 5 and Appendix A). At necropsy, the microbiota profiles appeared markedly different in composition compared to the post-FMT time point at both the phylum and family taxonomic levels. The most dramatic difference was the large apparent increase in the relative abundance of Firmicutes, which ranged from 91% to 96% in recipient mice at necropsy compared to 34%–57% post-FMT (Figure 5). A corresponding decrease in Bacteroidetes was evident, as the terminal microbiomes contained only 1.7% to 5% bacteria of this phylum. In addition, the apparent increase observed for Verrucomicrobia (species: *A. muciniphila*) observed at the post-FMT time point largely disappeared by 22 weeks, with only 0.6% to 3.8% of this phylum remaining. Conversely, Actinobacteria, which were present at very low abundance levels in post-FMT mice (<0.4%), were observed more frequently at necropsy (0.76% to 1.4%). Not only was a drastic change in the relative abundance of Firmicutes and Bacteroidetes evident at necropsy, major shifts in the dominant bacteria within the Firmicutes phylum were apparent. Of note, families Streptococcaceae, Turicibacteriaceae, Clostridiaceae, and unclassified members of the Clostridiales order were more abundant in recipient mice at necropsy compared to the post-FMT time point.

Similar comparisons were made for the terminal fecal microbiota profiles to determine whether FMT from lean or obese donors resulted in lasting changes to the recipient mouse microbiomes (Figure 5, Appendix A). Within the Firmicutes phylum, a few taxa were notably different between mice that received FMT from lean or obese donors. For mice fed the AIN diet, FMT from obese donors increased the abundance of Eubacteriaceae (genus: *Pseudoramibacter_Eubacterium*) by 13-fold compared to mice that received a transplant from lean donors (Appendix A). *Pseudoramibacter_Eubacterium* was also identified as a discriminating taxon for obese-FMT mice, most notably those fed both the AIN and TWD diets (Figure 7A,C). In addition, Mogibacteriaceae was more abundant in obese-FMT recipient mice than their lean counterparts by 2.8-, 2.3-, and 1.5-fold in mice fed AIN, DIO, or TWD diets (Appendix A), and this family discriminated obese-FMT recipient mice from lean recipients by LEfSE analysis for all diet groups (Figure 7). A similar trend was evident for *Enterococcus* (family: Enterococcaceae), which was 4.0- and 3.5-fold more abundant in obese-FMT recipient mice compared to lean recipients fed the DIO or TWD diets (Figure 5, Figure 7B,C, Appendix A). An opposite trend was evident for genus *Anaerotruncus* (family: Ruminococcaceae), which was measured in lean recipients at 10-, 4.4-, and 2.1-fold greater abundance than obese-FMT mice fed AIN, DIO, or TWD diets (Appendix A): *Anaerotruncus* was identified by LEfSe analysis to discriminate obese-FMT mice from lean-FMT mice for all diets (Figure 7). 

Although Firmicutes was the dominant phylum at the necropsy time point, other notable differences were evident for taxa in other phyla. However, a relatively low-abundance genus, *Collinsella*, discriminated lean-FMT from obese-FMT mice fed either the DIO or TWD diets (Figure 7B,C), likely due to the much greater relative abundance of *C. aerofaciens* in lean-FMT mice compared to their obese-FMT counterparts (Appendix A). Although the relative abundance of *Akkermansia* was notably lower at necropsy than at the post-FMT time point (Appendix A), this genus was identified as discriminating between lean mice fed the AIN diet (Figure 7A). Within the Bacteroidetes phylum, *Barnesiellaceae* was again associated with a significant effect of body type for mice fed the TWD (Figure 7C).

At the 22-week necropsy time point, we noted a number of taxa that were associated with specific diet treatments that appeared not related to either the lean or obese body type of the human donors (Appendix A). For example, Turicibacteraceae (genus: *Turicibacter*) was measured at 2.0- and 3.2-fold greater relative abundance in mice fed the TWD compared to those fed AIN or DIO diets, and this family was identified by LEfSE analysis as a discriminating taxon for mice fed the TWD compared to the AIN or DIO diets (Appendix A). *Lactococcus* (family: Streptococcaceae) was 47% or 58% more abundant in mice fed a DIO diet compared to those given AIN or TWD diets, and this genus was the most discriminating taxon (highest absolute LDA score) for the DIO diet compared to the other test diets (Appendix A). For the AIN diet group, *Allobaculum* (family: Erysipelotrichaceae) was noted as a discriminating genus compared to the DIO and TWD groups (Appendix A). *Allobaculum* was moderately abundant in AIN-fed mice at 5.9% on average compared to only 0.78% and <0.01% for mice fed DIO and TWD diets, respectively.

Correlation analyses were performed to determine whether the relative abundance of bacteria families present in fecal microbiota at necropsy was associated with phenotype measures independent of experimental categories. Strong positive correlations were identified for Clostridiaceae, Eubacteraceae, and Turicibacteraceae for final body weight, fat mass, and relative fat mass, and a corresponding negative correlation was observed for percentage lean mass (Figure 8). Alternatively, Enterococcaceae, Erysipelotrichaceae, and Lachnospiraceae were negatively correlated with fat mass and fat as percentage of body weight. Only two families were positively correlated with lean mass, including Turicibacteraceae and Eubacteriaceae, and only Turicibacteraceae was positively associated with glucose tolerance (Figure 8). 

### 3.7. Microbiome Profile Comparisons

Taxonomy data were analyzed using clustering, data reduction, and correlation methods to examine microbiome profiles associated with human donors and mouse recipients. First, hierarchical cluster analyses of the initial mouse microbiome and each of the six human donors revealed that the mouse microbiome was very distinct from that of any of the human donors (Appendix A). In addition, the microbiota profiles did not cluster separately according to the body type of the donor in either hierarchical cluster analysis or PCA. At the post-FMT time point, fecal microbiome profiles for recipient mice appeared to segregate according to donor body type more so than experimental diet (Figure 9). This trend was particularly notable in the hierarchical cluster analysis, for which most lean-FMT recipient mice were grouped within one major branch of the sample tree and the remainder were grouped in two other main branches (Figure 9C). However, though the recipient mouse microbiomes appeared to somewhat separate by donor body type, none of the mouse microbiota profiles co-clustered with the human donors (Figure 9B). Moreover, the correlation analyses showed that most of the recipient mouse microbiomes did not significantly correlate with the donors, and those that did had relatively high correlation *r* values in excess of 0.7 (Figure 9D). At the 22-week necropsy, a clear effect of experimental diet was observed for both the PCA and HCC clustering methods, as all mouse microbiota profiles segregated completely by diet group (Figure 10A,C), though none of the mouse microbiomes were similar to the human donors (Figure 10B). In addition, at this later point, the recipient mouse fecal microbiomes were substantially less similar to the human donor profiles, as exemplified by the overall lower Pearson *r* values and fewer significant correlations (Figure 10D).

### 3.8. Alpha and Beta Diversity of Recipient Mouse Fecal Microbiomes

Alpha diversity is a measure of the species representation within a biological sample and was determined for the recipient mouse microbiomes using several different metrics. At the post-FMT time point, significant differences in alpha diversity were evident when comparing recipient fecal microbiomes of mice fed the AIN and DIO diets (diet post hoc *p* < 0.05), including number of OTUs, Chao1 richness, and Faith’s phylogenetic diversity (Figure 11A–C), though no differences between the TWD diet and either the AIN or DIO diet was evident for these alpha diversity measures. Of additional note, no significant differences in these unweighted alpha diversity measures were noted when comparing lean- versus obese-FMT recipients. The Shannon index, which is weighted for species abundance, was significant for the DIO diet compared to both the AIN (diet post hoc *p* < 0.001) and the TWD (diet post hoc *p* < 0.05) diets, but microbiomes for mice fed the AIN and TWD diets were not different for this measure of alpha diversity (Figure 11D). In addition, a significant difference in the Shannon index was noted for mice fed the TWD and transplanted with mice from lean versus obese donors, with mice receiving lean-FMT having a higher Shannon index indicative of greater evenness and richness compared to their obese recipient counterparts. At necropsy, more significant effects of the experimental diet were noted for alpha diversity as measured by OTUs, Chao1 richness, or Faith’s phylogenetic diversity, with mice fed the DIO diet having significantly different scores compared to those fed AIN or TWD diets (Figure 11E,F). In addition, more differences in alpha diversity were noted at necropsy when comparing lean- versus obese-FMT recipient mice, with significant differences evident for OTUs and Faith’s phylogenetic diversity for the AIN and TWD diets (Figure 11E,G) and for Chao1 richness for mice fed the AIN diet (Figure 11F). Overall, Shannon index values were greater in mice at the necropsy time point, and this index was significantly different only between mice fed the DIO and TWD diets (diet post hoc *p* < 0.01). Interestingly, Shannon index values were significantly higher in obese-FMT recipient mice compared to their lean counterparts for mice fed the AIN and DIO diets, but not the TWD (Figure 11H).

Beta diversity unweighted and weighted unifrac distance matrices were visualized as PCoA plots for the effects of donor body type and experimental diet on diversity of the recipient mouse microbiome. At the post-FMT time point, unweighted unifrac distances were highly segregated by human donor (permanova *p* = 0.001), with an apparent cluster of lean donors and human donor O5 that were clearly distinct from the obese donors O4 and O6 (Appendix A). Some clustering of lean donors was also evident for weighted unifrac distances for lean donors, though the data were more scattered overall (Appendix A). At necropsy, unweighted unifrac distances were somewhat separated by donor body type (permanova *p* = 0.001) (Appendix A), though more dispersion was apparent compared to the post-FMT time point. Weighted unifrac distances were much less organized by body type, though a significant difference was still apparent (permanova *p* = 0.013). When categorizing the unifrac distances by experimental diet, some separation according to diet was apparent for weighted distances post-FMT (permanova *p* = 0.001) (Appendix A), and clear separation for the TWD and DIO diets was evident for weighted distances at necropsy (Appendix A).

Beta diversity was also assessed for unifrac distance matrices calculated between lean- and obese-FMT recipient fecal microbiomes for each diet group (Figure 12). For all diets, at the post-FMT time point, the unweighted distances were highly correlated with the human donor, and some separation by body type was apparent, though donor O5 tended to cluster with the lean donors (Figure 11A). Unweighted distances were more dispersed for all diets, yet beta diversity was significantly different for lean versus obese recipient mice for each diet group (permanova *p* = 0.001). At necropsy, unweighted distances were highly dispersed (permanova *p* = 0.001), with separation by body type most apparent for mice fed the AIN diet (Figure 12B). However, weighted unifrac distances between lean and obese recipient mice were not significantly different for any of the test diets.

## 4. Discussion

Obesity and related metabolic disorders are a growing threat to worldwide public health. Interventions to alleviate this trend are needed and are the subject of numerous investigations across scientific disciplines. Emerging literature has suggested that the gut microbiome may be involved in the etiology of obesity. Therefore, interventions that modify the gut microbiome, such as FMT, may be viable options in combatting obesity. In the current study, we examined whether mice humanized with fecal microbiota from obese or lean human donors would have different metabolic phenotypes when fed obesogenic or control diets. Contrary to our hypothesis, FMT from obese or lean donors did not significantly affect final body weight or body composition in recipient mice fed any of the experimental diets. This result conflicts with reports from earlier studies in which the transfer of microbiota from obese mice led to increased weight gain in recipients [15,46] or the serial transfer of microbiota from obese human donors to mice followed by a second mouse-to-mouse transfer increased weight gain in recipient mice [16]. Immediately following FMT from human donors, the fecal microbiome profiles for recipient mice appeared to be more influenced by the human donor than the experimental diet, though the recipient mice bacteria profiles were clearly distinct from those of the original human donors. By the end of the study, the fecal microbiome profiles for recipient mice were clearly strongly influenced by the experimental diet, though a number of taxa that discriminated between lean and human donors were identified at this late time point. Overall, these results suggest that the basal diet was the most important factor in shaping the gut microbiome composition.

As expected, mice fed the high-fat DIO diet were significantly heavier, had a higher fat mass, and had impaired glucose tolerance relative to mice fed the other diets. These observations were not surprising, as the DIO diet fed to the obesity-prone C57BL/6 mouse strain is a widely used model for obesity and associated metabolic perturbations [47,48]. Despite its greater energy density compared to the AIN diet, consumption of the TWD did not significantly increase body weight gain, increase fasting glucose level, or impair glucose tolerance relative to mice fed the AIN diet, although mice fed the TWD did have moderately increased fat mass relative to body weight. Similarly, Monsanto et al. reported that mice fed the AIN and TWD diets had a healthier metabolic phenotype compared to mice fed the DIO diet [37]. In this study, there was no interaction between diet and the microbiota donor source at any of the endpoints related to obesity. This finding was contrary to our hypothesis that microbiota from obese human donors might exacerbate diet-driven increases in obesity and changes in metabolism.

The initial mouse microbiome had a large community of S24-7 bacteria, which was completely depleted after the FMT procedure. The S24-7 family of bacteria (order: Bacteroidales) is common in the murine microbiome and increased in relative abundance in response to prebiotic supplementation [49]. Interestingly, the S24-7 family was the top discriminating feature of the intestinal microbiome of uncontacted Yanomami Amerindians in relation to other human groups, including Americans [50], suggesting that this family may be especially sensitive to antibiotic exposure. Conversely, *Prevotella* from human donors was not established in the mouse microbiome regardless of its relative abundance in the human donors’ microbiota profiles. This observation indicates that the mouse gastrointestinal tract may be ill-suited to support the growth of this genus, which has been suggested previously [10]. The loss of one or more key bacteria taxa following FMT is not surprising, as it has also been shown that the community of bacteria that is established after transplant is more likely to persist if those species are endemic to the mouse gut [51]. Moreover, by the end of the feeding period, there was an almost complete depletion of Bacteroidetes independent of body type of the human donor or experimental diet. Bacteroidetes was present in the human donor samples, in the initial mice microbiomes, and in the recipient mice post-FMT. However, by the end of the 22-week study, most bacteria within the Bacteroidetes phylum were replaced with Firmicutes. The lack of an established Bacteroidetes community after the human FMT differed from past studies [11,52]. 

LefSe analysis is a useful tool that defines taxa that discriminate between experimental groups and allows for direct comparisons to previous microbiome research in both mice and humans. Following FMT, *Holdemania, Coprococcus*, *Ruminococcus,* and *C. aerofaciens* were all identified as taxa that consistently discriminated between mice receiving FMT from obese or lean human donors regardless of the experimental diet. At the post-FMT time point, other discriminating taxa for obese FMT recipients were identified in mice fed one or more of the test diets, including *A. muciniphila*, *C. aerofaciens*, Erysipelotrichales, and Barnesiellaceae. Some of these taxa have been linked to obesity or metabolic syndrome in other studies in humans or animal models. For example, Le Chatelier et al. observed a high abundance of *A. muciniphila* in human microbiomes with high microbial diversity [9]. In addition, *A. muciniphila* has also been shown to be less abundant in obese and diabetic mice compared to controls [53] and was more abundant in patients after gastric bypass surgery [54], suggesting that *A. muciniphila* may be associated with leanness in both mice and humans. Although *C. aerofaciens* was associated with the lean microbiomes in our study, patients with pre-diabetes and type 2 diabetes have been found to have microbiomes enriched with *C. aerofaciens* relative to healthy controls [55]. Alternatively, in the present study, *Blautia* was more abundant in lean-FMT recipient mice fed either AIN or DIO diets compared to their obese FMT counterparts, though this difference was not persistent to the end of the study at 22 weeks. The literature concerning *Blautia* and obesity or metabolic syndrome has been equivocal. In one report, obese or overweight humans were fed a low-calorie, high-protein diet for three weeks to induce weight loss: Following this dietary intervention, researchers observed an increase in the relative abundance of *Blautia* in fecal samples compared to baseline values [56]. Alternatively, *Blautia* has been found to be positively correlated with higher body weight in C57BL/6J mice fed either high- or low-fat diets [57]. A similar result was found in another mouse study, in which *Blautia* was less abundant in mice with a lower body weight triggered by consumption of tea polyphenols [58]. In order to obtain more clarity in this area of microbiome research for future studies, mice could be reconstituted with “simplified intestinal microbiota”, which would allow for the study of how diet affects microbe–diet interaction and possibly the host’s metabolic phenotype [59].

At necropsy, discriminating taxa for obese-FMT recipients were identified, including Mogibacteriaceae and *Anaerotruncus*, which were consistently associated with obese-FMT recipients fed all experimental diets. Other features that discriminated between mice receiving bacteria from obese donors fed one or more of the experimental diets included *Holdemania*, Clostridiaceae, Christensenellaceae, *Pseudoramibacter_Eubacterium*, and *Ruminococcus*. Past studies have observed that the majority of Firmicutes in obese mice belonged to the order Clostridia, specifically Clostridium cluster XIVa, which is known for butyrate production [60]. Alternatively, past studies have shown an increased abundance of Christensenellaceae in subjects with normal BMI (under 25) [61]. *Coprococcus* was reduced in patients with nonalcoholic fatty liver disease, which is associated with metabolic syndrome, and this association was independent of BMI or insulin resistance [62]. Mogibacteriaceae is typically associated with a lean phenotype, as was shown in a study of mice that lost weight as a consequence of metabolic suppression through cold exposure [63]. This family was also enriched in mice fed a low-fat diet compared to *db/db* mice and or mice fed a high-fat diet [64]. Mogibacteriaceae was also observed to be associated with leanness in a healthy Japanese population with BMI under 25 compared to individuals with a BMI greater than 30 [65]. Less information is available for *Anaerotruncus,* though one group reported that this genus was enriched in a cohort of lean Chinese children compared to their obese counterparts [66].

The selections of the animal model and the basal or experimental diets are important considerations when designing preclinical studies that incorporate analysis of the gut microbiome. In the present study, we observed that the composition of the gut microbiome was influenced to a greater extent by the experimental diet compared to the body type of the human microbiota donor. However, emulating human dietary patterns in animal models can be problematic due to differences in the nutrient composition of standard animal diets compared to typical human nutrient intakes [28]. The DIO diet has been used previously to evaluate the Western dietary pattern in microbiome studies [46,52]. Although the DIO diet reliably produces an obese phenotype in genetically susceptible mice, it has little relevance to the Western dietary pattern with respect to its micronutrient composition and its lipid profile. In this study, we compared changes in phenotype biomarkers of obesity and metabolic syndrome and changes in the gut microbiome in mice fed the DIO diet versus the TWD diet, the latter of which was formulated to more closely reflect the average micro- and macronutrient intake of Americans. 

Although a comparison between obese-FMT recipients and their lean counterparts was the primary focus of this study, we also examined mouse microbiome profiles to identify possible discriminating taxa for the experimental diets. LEfSe analyses revealed some interesting discriminating taxa for AIN and TWD diets, which were largely without effect for the obese- or metabolic syndrome-related endpoints, including members of the Bacteroidetes phylum, such as Bacteroidetes, Barnesiellaceae, Odoribacteraceae, and Rikenellaceae. Others have also examined the contribution of high-fat or purported “Western”-type diets on the composition of the gut microbiome in mice. Turnbaugh et al. humanized germ-free mice with bacteria from a human donor and then fed the recipient mice either a chow diet or a high-fat, high-sugar diet [67]. They then performed mouse-to-mouse FMT and examined the composition of the gut microbiome in the second cohort of recipient mice, which were also fed either the chow or high-fat, high-sugar diets. Regardless of the body type of the mouse donors, the second cohort of mice fed high-fat, high-sugar diets had a decreased abundance of Bacteroidetes [67]. In our study, Bacilli was identified as discriminating between mice fed the DIO diet following FMT, though this was not consistent at the end of the study. These results are similar to an investigation by Turnbaugh et al., as they also observed an increase in the abundance of Bacilli in mice fed a DIO diet [52]. 

A limitation of many preclinical animal studies, including the present study, that examine the effects of diet on the microbiome is the use of purified diets. Commercial, purified mouse diets contain a limited selection of ingredients, such as casein, corn starch, sucrose, and a vegetable oil, and use a single-fiber source, cellulose. While very useful for the replication of animal model nutrition studies, these purified diets lack the complex bioactives and diverse fiber sources present in whole foods. Dalby et al. compared the microbiomes of mice fed either a low-fat chow diet, a low-fat purified diet, or a high-fat purified diet [68]. They reported that the fat content of the diet was the primary driver of obesity despite the observation that mice fed low-fat purified and low-fat chow diets had distinct microbiomes, including a different Bacteroidetes/Firmicutes ratio [68]. The authors urged caution in interpreting the results of early microbiome studies that compared refined DIO diets to control chow diets, as the phenotypes of these mice were essentially uncoupled from microbiota composition because of the outsized effects of the unrefined chow diet on the microbiome. In our study, the lack of a complex food matrix, including diverse sources of soluble fiber, may have contributed to large-scale changes in the microbiome composition when comparing bacteria profiles obtained at the end of FMT to profiles after 18 weeks of dietary treatment. To address this limitation of refined diets, future studies could employ a diet that emulates the complex food matrix and dietary fiber profile of human diets.

The overall gut microbiota profiles for our human donors did not segregate neatly by donor body type. This observation was consistent with other studies and reinforces the concept that other factors, such as diet, may be critically important in shaping the microbiome [18]. For example, Turnbaugh et al. investigated the relationship of diet, gut microbiota, and energy balance in germ-free mice conventionalized with microbiota from obese or lean mice [46]. They reported that the observed shift in the bacteria population was dependent on consumption of the DIO diet but independent of the body type of the mouse FMT donor, indicating that diet can overtake donor effects. Our results align with the concept of diet having a dominant role in determining microbiome composition, as we saw a major shift determined by diet exposure, as observed in other studies [18,46,68].

Recent meta-analyses have challenged the concept of whether obesity is typified by a specific signature microbiome. For example, some studies have suggested that the obese gut microbiome is typified by a higher ratio of Firmicutes/Bacteroidetes [11,15,52,60] as well as lower alpha diversity compared to lean individuals [9,14]. However, meta-analyses of human studies have failed to find higher Firmicute/Bacteroidetes ratios in obese versus lean subjects and, contrary to early animal studies, increased alpha diversity in obese humans [13,69]. Moreover, these meta-analyses have not identified individual taxa or overall phylotypes associated with obesity, indicating that the relationship between microbiota composition and obesity is not straightforward.

This study design did have several limitations. As noted above, the transfer of human bacteria to mouse recipients was not complete, as some taxa present in humans were not successfully established in mice (e.g., *Prevotella*). This deficiency is not unexpected in human-to-mouse FMT, although our prior work using this broad-scope antibiotic method showed that about 70% of the donor sequence mass was recovered in recipient mice [30]. The lack of a transfer of an obese phenotype from donors to mice could also have been related to our FMT model. It is possible that obesogenic bacteria are more easily transmitted and engrafted in gnotobiotic mice compared to antibiotic-treated mice. However, the FMT approach used in this study is more relatable to the application of FMT for human patients in a clinical setting. Another limitation was the small pool of human donors available from which to select representative lean or obese individuals. The bacteria profile for each of the donors was quite distinct at the family taxonomic level, with no apparent segregation by donor body type for the overall bacteria profiles. In addition, the limited sample size of three donors for each body type likely reduced the statistical power for identifying discriminating bacteria in mice that received lean or obese FMT. It is important to note that the human donors were selected based on phenotype parameters (BMI and waist circumference), not microbiome profiles. In future studies, a larger population of human donors could be screened and selected based on both phenotypic metrics and microbiome profiles to ensure greater homogeneity among donors within each experimental group. As noted above, the ingredients selected in formulating a rodent diet likely influence the gut microbiome by providing a diverse array of bioactive food compounds and fermentable fibers that may function as prebiotics. Future studies could address this limitation by constructing animal diets using whole-food sources with macro- and micronutrient compositions matching the purified formulations, although such diets themselves could suffer with respect to reproducibility. Lastly, the current study explored the gut microbiome composition using 16S rRNA sequencing data to determine the taxonomic classification of bacteria present in fecal samples. This approach tells us what taxa are present, but does not convey information about their metabolic function. It is possible that some taxa identified as discriminating between either lean- or obese-FMT or one of the experimental diets are functionally redundant: Their various functions may be covered by other bacteria that were not identified as differentially abundant. A metagenomics approach that predicts bacteria function by gene ontology classification could provide more extensive information about the putative function of specific bacteria within the gut microbiome ecosystem [70]. 

## 5. Conclusions

Our results and those of other investigators have indicated that the relationship between obesity and the composition of the gut microbiome is complicated. One must recognize that the etiology of obesity is multifactorial and that the condition is associated with many comorbidities, including impaired glucose metabolism and chronic inflammation [52,71,72]. On the other hand, the connection between diet and the gut microbiome is more straightforward, as the food consumed by the host can support homeostasis or promote dysbiosis. In this study, we observed that the microbiome of recipient mice was changeable by FMT from human donors, but that with continued feeding, diet was the overriding variable in shaping the gut microbiota population and metabolic phenotypes. Although our findings did not support our hypothesis, we believe these results are important because they suggest that microbiota transfer may not be a useful therapeutic option for obesity without a concurrent change in diet. Interestingly, a research group in the Netherlands used fecal transfer from lean, healthy volunteers to treat patients with obesity and metabolic syndrome in two separate studies [19,20]. In both studies, glucose metabolism was improved by FMT from lean donors, but obese recipients did not lose weight, suggesting that microbiota transfer from a lean donor was insufficient to change the obese phenotype in recipients. In conclusion, the results of this study investigating the interaction between a fecal microbiome transfer from lean or obese human donors and the basal diet fed to recipient mice demonstrated the critical role of diet in determining the gut microbiome composition, which is in agreement with other studies [18,27,67]. Moreover, in this 22-week feeding study, fecal microbiota transfer from obese human donors did not confer host traits to recipient mice. While specific taxa were identified that discriminated between mice that received an FMT from either lean or obese human donors, the overall gut bacteria profiles were much more reflective of the diet consumed.

## Figures and Tables

**Figure 1 nutrients-11-01630-f001:**
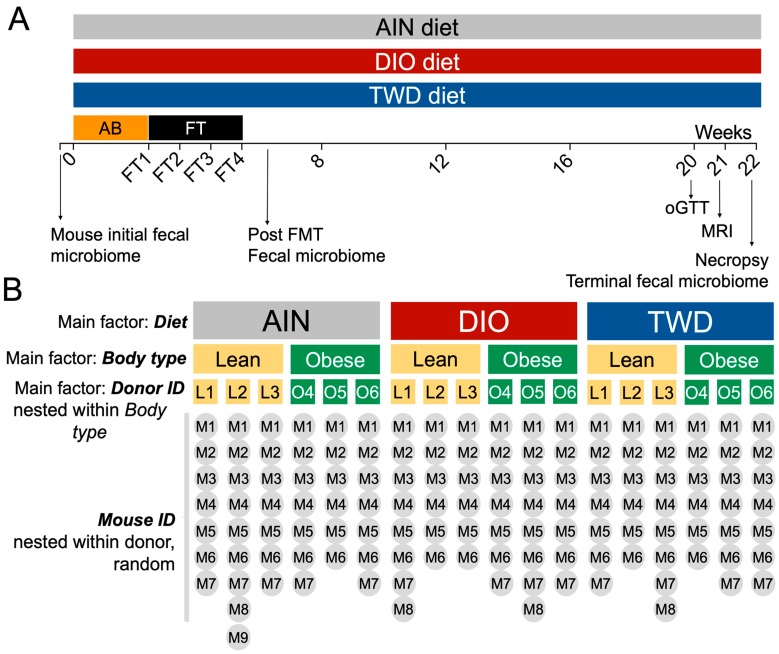
Experiment design and allocation of mice to experimental diets and human donor groups. (**A**) Experiment timeline indicating the timing of antibiotic treatment (AB), fecal transfer from human donors, experimental diets, collection of fecal material for microbiome sequencing and other endpoints. (**B**) Experiment design, including main factors included in the basal diet (AIN93G (AIN), diet-induced obesity (DIO), or total Western diet (TWD)), body type (lean or obese), and human donor ID (L1, L2, L3, O4, O5, O6) nested within the body type. Mouse ID is a random factor nested within both donor and body type.

**Figure 2 nutrients-11-01630-f002:**
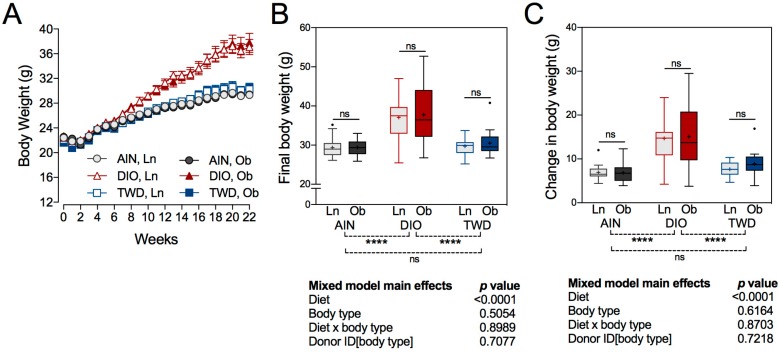
Body weight gain. (**A**) Body weight gain over the 22-week study period. Data shown are the mean ± standard error of measurement (SEM) for each group. (**B**) Final body weight and (**C**) change in body weight data are presented as Tukey box plots (box, 25th to 75th percentiles; whiskers, 1.5 interquartile range (IQR); +, mean) (*n* = 6 to 9). The table to the right of this panel shows *p*-values for the main effects of each experimental factor as determined by the mixed model analysis. Below the plot, brackets indicate the results of Tukey post hoc tests for the overall effects of each diet. Within each diet group, symbols above the box and whisker bars indicate the results of post hoc tests comparing mice that received bacteria from lean (Ln) or obese (Ob) human donors. **** *p* < 0.0001, ns = not significant.

**Figure 3 nutrients-11-01630-f003:**
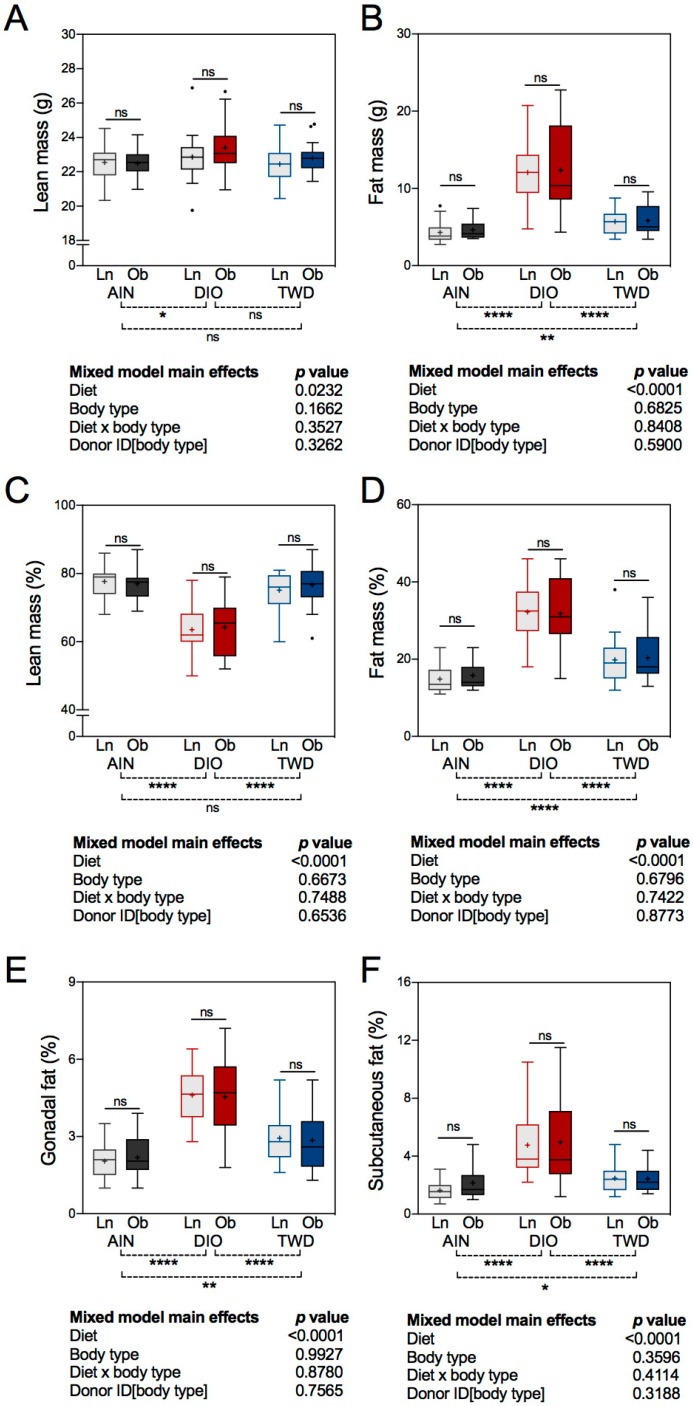
Body composition and fat distribution. Data for lean mass (**A**), fat mass (**B**), lean mass as percent of body weight (**C**), fat mass as percent of body weight (**D**), gonadal fat pad as percent of body weight (**E**), and subcutaneous fat as percent of body weight (**F**) are shown in Tukey box plots (box, 25th to 75th percentiles; whiskers, 1.5 IQR; +, mean) (*n* = 6 to 9). The tables below each panel show *p*-values for the main effects of each experimental factor as determined by the mixed model analysis. Below each plot, brackets indicate the results of Tukey post hoc tests for the overall effects of each diet. Within each diet group, symbols above the box and whisker bars indicate the results of post hoc tests comparing mice that received bacteria from lean (Ln) or obese (Ob) human donors. * *p* < 0.05, ** *p* < 0.01, **** *p* < 0.0001, ns = not significant.

**Figure 4 nutrients-11-01630-f004:**
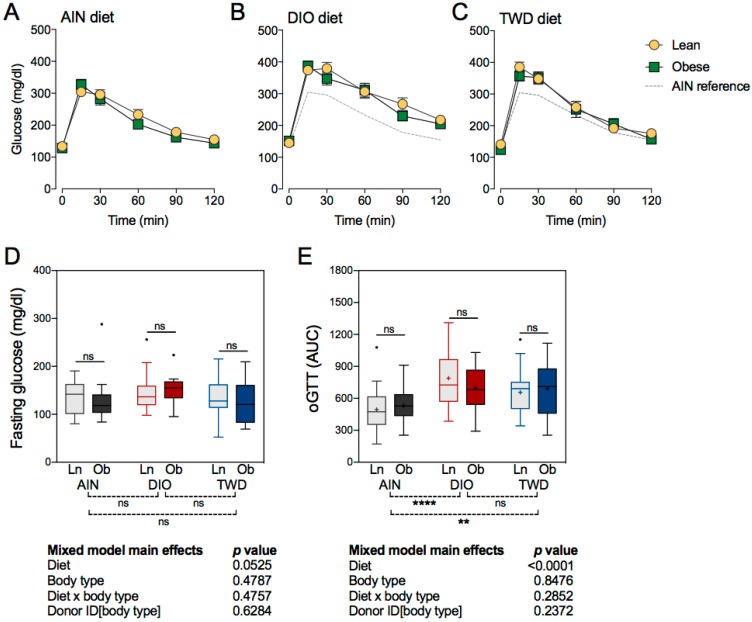
Oral glucose tolerance test (oGTT). Glucose tolerance was assessed by calculating the area under the curve (AUC), with a baseline set at 100 mg/dl, for mice that received a fecal transfer from either lean (Ln) or obese (Ob) human donors. Data shown are mean ± SEM (*n* = 6 to 9) for blood glucose concentrations with respect to time following an oral glucose dose of 10 mg glucose/kg body weight in mice fed an AIN diet (**A**), DIO diet (**B**), or TWD diet (**C**). Fasting blood glucose (**D**) and oGTT AUC (**E**) values are shown as Tukey box plots (box, 25th to 75th percentiles; whiskers, 1.5 IQR; +, mean) (*n* = 6 to 9). The tables below each panel show *p*-values for the main effects of each experimental factor as determined by the mixed model analysis. Below each plot, brackets indicate the results of Tukey post hoc tests for the overall effects of each diet. Within each diet group, symbols above the box and whisker bars indicate the results of post hoc tests comparing mice that received bacteria from Ln or Ob human donors. ** *p* < 0.01, **** *p* < 0.0001, ns = not significant.

**Figure 5 nutrients-11-01630-f005:**
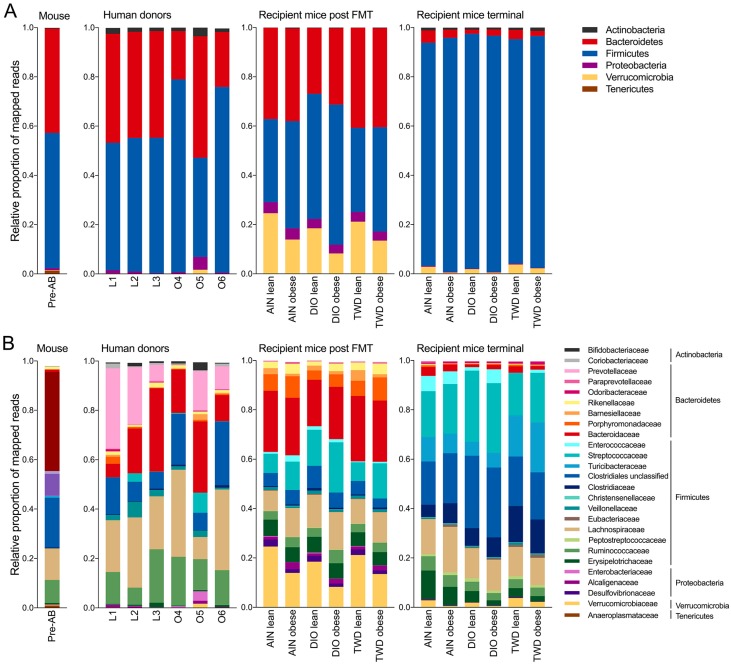
Classification of human or mouse fecal bacteria by phyla or family. Data shown are the relative abundance of bacteria to annotated phyla (**A**) or family (**B**) for the initial mouse microbiome (pre-AB), the original human donors, the recipient mice after the last fecal transfer from human donors, or the recipient mice at the end of the study. Taxonomic classifications for individual recipient mice post-fecal microbiome transfer (FMT) and necropsy (terminal) are provided in Appendix A. Abbreviation: AB, antibiotic.

**Figure 6 nutrients-11-01630-f006:**
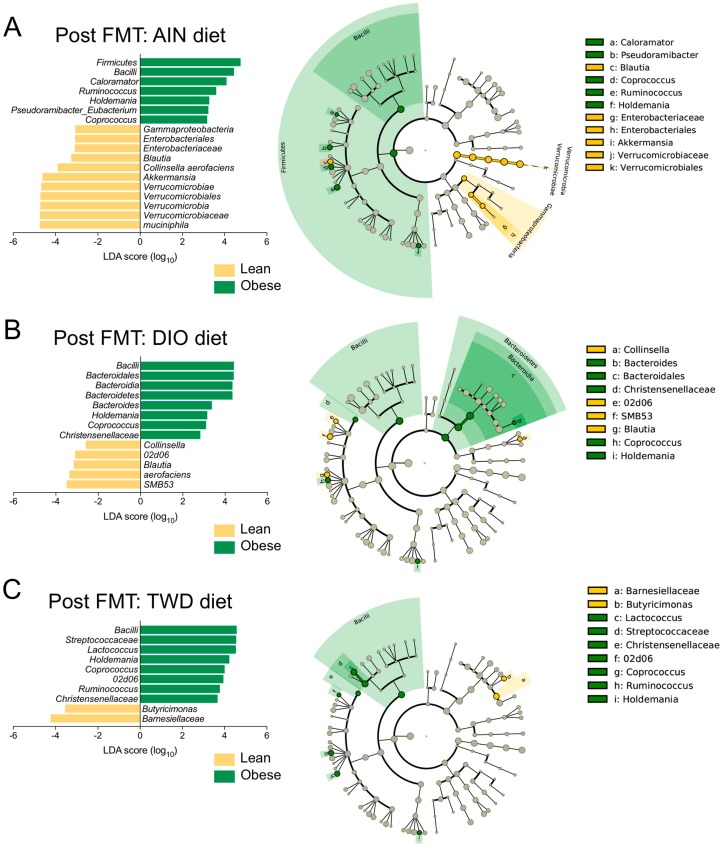
Taxa that discriminated between lean and obese human donors in recipient mice post-fecal transfer. Linear discriminant analysis with effect size (LEfSe) analyses were performed using relative abundance data averaged by human donor at the highest classification level taxonomy available. Data shown are the log_10_ linear discriminant analysis (LDA) scores following LEfSe analyses and the hierarch of discriminating taxa visualized as cladograms for class comparisons between lean human donors and obese human donors in recipient mice fed (**A**) AIN, (**B**) DIO, or (**C**) TWD diets. Relative abundance data of selected taxa of interest are shown in Appendix A.

**Figure 7 nutrients-11-01630-f007:**
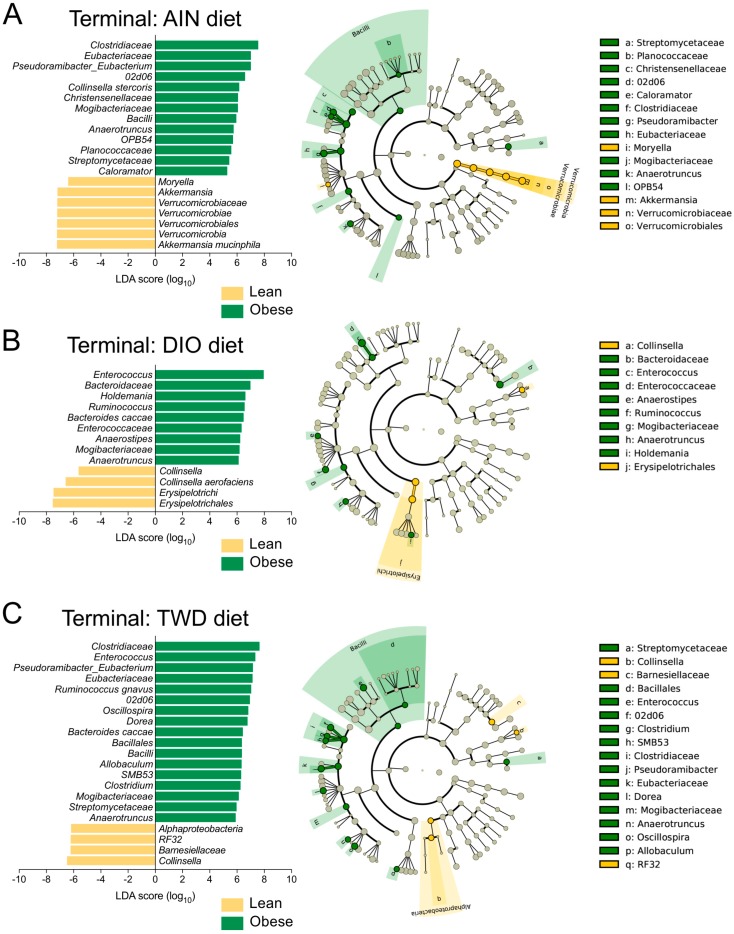
Taxa that discriminated between lean and obese human donors in recipient mice at necropsy. LEfSe analyses were performed using relative abundance data averaged by human donor at the highest classification level taxonomy available. Data shown are the log_10_ linear discriminant analysis (LDA) scores following LEfSe analyses and the hierarch of discriminating taxa visualized as cladograms for class comparisons between lean human donors and obese human donors in recipient mice fed (**A**) AIN, (**B**) DIO, or (**C**) TWD diets. Relative abundance data of selected taxa of interest are shown in Appendix A.

**Figure 8 nutrients-11-01630-f008:**
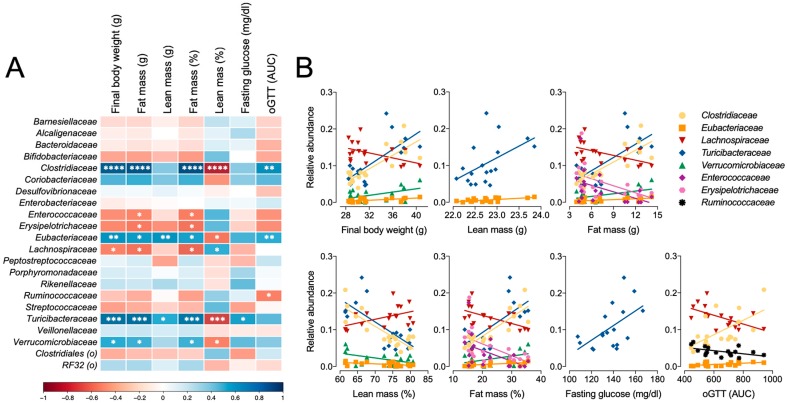
(**A**) Correlation analysis for major bacteria families with key phenotype endpoints. (**B**) Linear regressions are shown for bacteria significantly correlated for each endpoint. The scale indicates Pearson r correlation value and direction. Pearson correlation significance indicated by * *p* < 0.05, ** *p* < 0.01, *** *p* < 0.001, **** *p* < 0.0001.

**Figure 9 nutrients-11-01630-f009:**
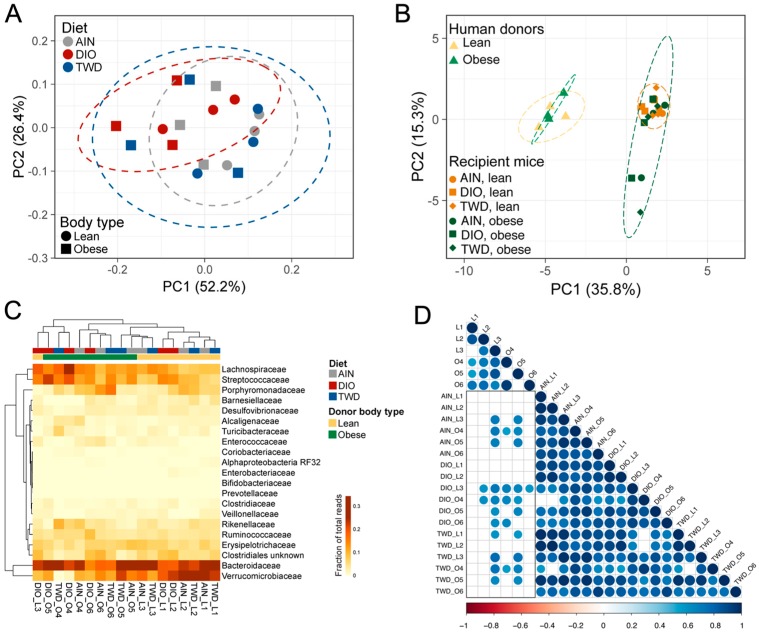
Hierarchical clustering and principal components analysis of recipient mouse microbiomes post-FMT. (**A**,**B**) Principal components analysis (PCA) of family-level taxonomy data plotted according to the first two principal components for FMT recipient mice (A), comparing recipient mice to original human donors (B). Variation attributed to each component is shown. Dashed lines represent the 95% confidence ellipse. (**C**) Unsupervised, bidirectional hierarchical cluster analysis of taxonomy relative abundance data for bacteria families comprising at least 1% of the fecal microbiome from all mice prior to antibiotic treatment (pre-AB) and from each of the six human donors (lean: L1, L2, L3; obese: O4, O5, O6). Clustering was performed using Euclidian distance with average linkage. Heatmap color scale indicates relative abundance as fraction of total reads. (**D**) Pearson correlation plot showing correlation between human donor microbiomes and recipient mice. The color scale indicates the size and direction of the correlation coefficient. Only significant (*p* < 0.05) correlations are shown. The black box highlights the correlation of FMT recipient mice with their human donors.

**Figure 10 nutrients-11-01630-f010:**
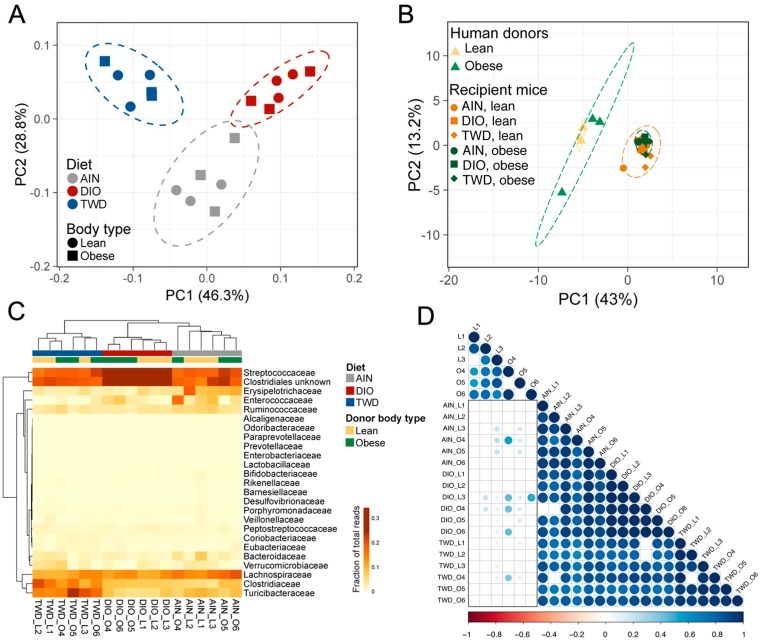
Hierarchical clustering and principal components analysis of recipient mouse microbiomes at necropsy. (**A**–**B**) PCA of family-level taxonomy data plotted according to the first two principal components for FMT recipient mice (**A**), comparing recipient mice to original human donors (**B**). Variation attributed to each component is shown. Dashed lines represent the 95% confidence ellipse. (**C**) Unsupervised, bidirectional hierarchical cluster analysis of taxonomy relative abundance data for bacteria families comprising at least 1% of the fecal microbiome from all mice prior to antibiotic treatment (pre-AB) and from each of the six human donors (lean: L1, L2, L3; obese: O4, O5, O6). Clustering was performed using Euclidian distance with average linkage. Heatmap color scale indicates relative abundance as fraction of total reads. (**D**) Pearson correlation plot showing correlation between human donor microbiomes and recipient mice. The color scale indicates the size and direction of the correlation coefficient. Only significant (*p* < 0.05) correlations are shown. The black box highlights the correlation of FMT recipient mice with their human donors.

**Figure 11 nutrients-11-01630-f011:**
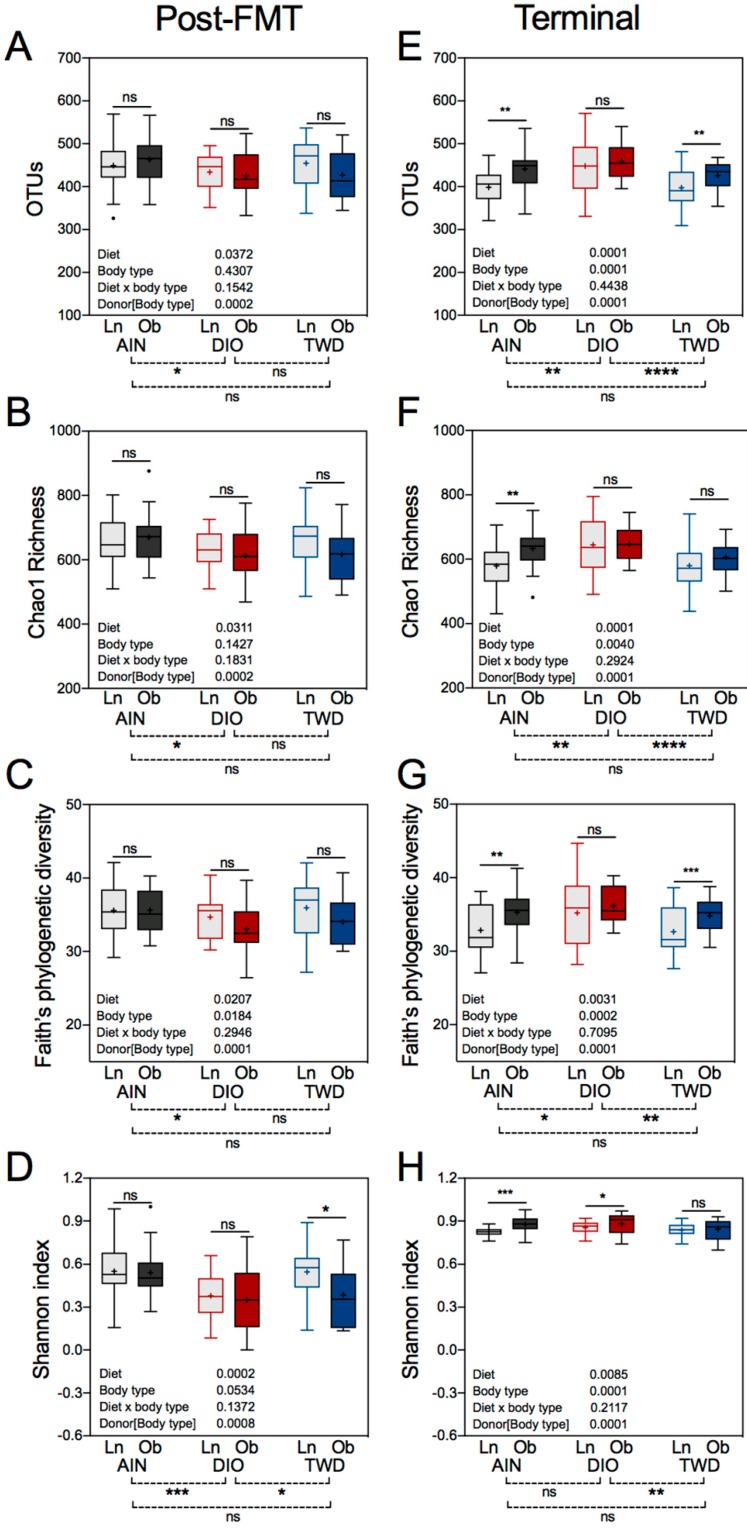
Alpha diversity of recipient microbiomes post-FMT and at necropsy. Operational taxonomical units (OTUs) (**A**,**E**), Chao1 richness (**B**,**F**), Faith’s phylogenetic diversity (**C**,**G**), and Shannon index (**D**,**H**) alpha diversity measures for fecal mouse microbiomes post-FMT and at necropsy are presented as Tukey box plots (box, 25th to 75th percentiles; whiskers, 1.5 IQR; +, mean) (*n* = 6 to 9). The tables within each panel show *p*-values for the main effects of each experimental factor as determined by the mixed model analysis. Below each plot, brackets indicate the results of Tukey post hoc tests for the overall effects of each diet. Within each diet group, symbols above the box and whisker bars indicate the results of post hoc tests comparing mice that received bacteria from lean (Ln) or obese (Ob) human donors. ** p* < 0.05, *** p* < 0.01, **** p* < 0.001, ***** p <* 0.0001, ns = not significant.

**Figure 12 nutrients-11-01630-f012:**
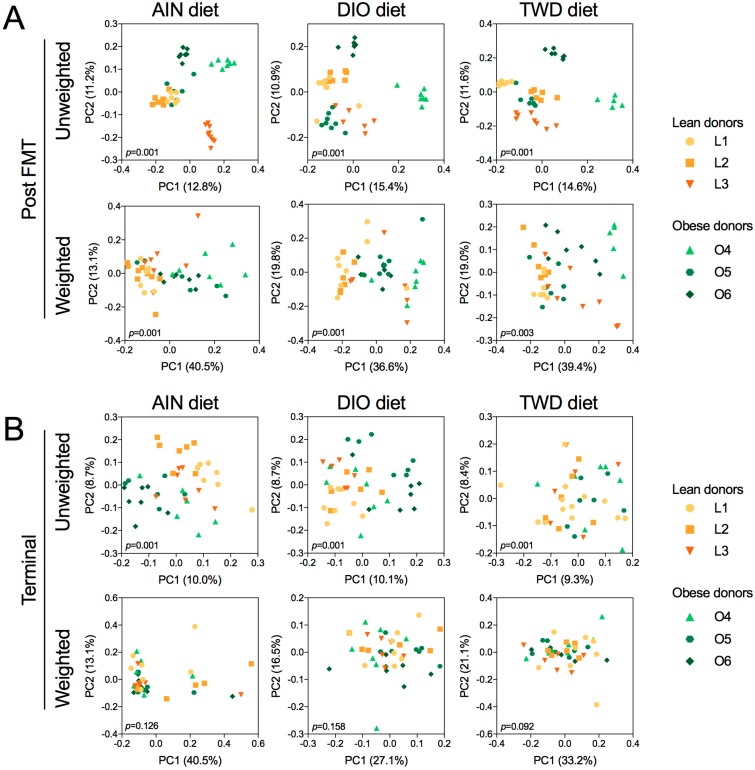
Beta diversity for post-FMT and terminal recipient mouse microbiomes. Data shown are the first two principal coordinates of unweighted or weighted unifrac distances for the post-FMT microbiome (**A**), and the terminal fecal microbiome (**B**). Plots are categorized by human donor body type within each experimental diet. Variation attributed to each coordinate is shown. Permanova *p*-values are shown for each plot.

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
