# Peer review of "Basal Diet Determined Long-Term Composition of the Gut Microbiome and Mouse Phenotype to a Greater Extent than Fecal Microbiome Transfer from Lean or Obese Human Donors"

_nutrients, 2019, doi:10.3390/nu11071630_

Round 1
Reviewer 1 Report
In the US obesity is a national epidemic and an healthcare emergency. This study set out to test the effect of gut microbial association and diet type.
This study, though complex, is well designed and neatly executed. It is intuitive knowledge that the amount of food intake directly impacts the body weight. Added to this, genetic make-up, current lifestyle, the overall portion size, foods loaded with empty calories are a few of the contributing factors to this health crisis.
The association of gut microbiome to overall health is well known, however, the purported hypothesis linking diet and the gut microbiome is interesting. A different perspective in this direction could had been to examine the association between the gut microbiome, micronutrient absorption, and overall health.
There are a few points that need clarifications:
Materials & Methods:
The treatment was scheduled for 22 weeks.
Please explain the rationale behind this 22 weeks trial?
FMT:
Fecal microbiome collection. I presume both distinct and collective fecal microbiome concentration (numbers NOT the biodiversity) might vary from sample to sample. The authors explained that the efforts were made to collect the fecal matter at a defined time of the day. However, they did not describe any attempt to quantify the fecal microbiome concentration. Instead 1g/ml fecal suspension was gavaged.
The fecal samples were frozen till they were deployed. The authors must explain the survival and revival of the microbiome during the freeze thaw.
May be my ignorance, but, it is my understanding that normal microbes will not survive the gastric environment. But the results demonstrated the microbes indeed survived the gastric environment. However, did the gastric environment alter the population composition? An explanation will add value to this study.
The water contained 1g/ml Amp. Did this antibiotic regimen continue through out the 22 weeks period. It must be explicitly stated under M&M section.
Statistical Analyses:
I am not qualified to comment on the statistical data analyses methodology. However, I am knowledgeable to understand the data. The following points need clarifications.
How the number of animals needed for each group was selected/arrived at?
For the genomic profiling, did they pool the fecal samples?
Results:
The results and the subsequent discussion sections are clearly presented.
Line 316: Body weight gain:
Is it body weight or body weight gain (rate). The animals lost weight during the first two weeks of the 22 wk tx period (figure 2A). But what happened during the native microbiome depletion period? Did the animals loose weight? So, this body weight loss is from the start of FMT or the start of the Ab treatment?
Is 12 hours post antibiotic withdrawal provide enough time to get the body rid off the antibiotics? How did this treatment affect the fecal microbiome establishment? I believe gastric environment combined with this short (12h) gap may be too much of an assault on the fecal microbiome population.
Conclusion: Line 856: Although are hypothesis- May be a typo?
This line states 'hypothesis was incorrect'. I believe the authors meant to state the experimental data disproved our hypothesis or did not support our hypothesis. Just a thought.
Author Response
Response to Reviewer 1
In the US obesity is a national epidemic and an healthcare emergency. This study set out to test the effect of gut microbial association and diet type.
This study, though complex, is well designed and neatly executed. It is intuitive knowledge that the amount of food intake directly impacts the body weight. Added to this, genetic make-up, current lifestyle, the overall portion size, foods loaded with empty calories are a few of the contributing factors to this health crisis.
The association of gut microbiome to overall health is well known, however, the purported hypothesis linking diet and the gut microbiome is interesting. A different perspective in this direction could had been to examine the association between the gut microbiome, micronutrient absorption, and overall health.
There are a few points that need clarifications:
Materials & Methods:
The treatment was scheduled for 22 weeks. Please explain the rationale behind this 22 weeks trial?
The goal of the experiment was to emulate FMT in humans and determine long-term effects of different dietary exposures. To our knowledge, this is the longest study of this nature. In our experience, we have reported adiposity differences between mice fed DIO diets and our TWD at approximately 14 weeks [1]. Therefore, by 22 weeks we would be confident of diet effects on phenotype and investigate long-term effects of different dietary exposures on the microbiome after FMT. This has been clarified in the text.
FMT:
Fecal microbiome collection. I presume both distinct and collective fecal microbiome concentration (numbers NOT the biodiversity) might vary from sample to sample. The authors explained that the efforts were made to collect the fecal matter at a defined time of the day. However, they did not describe any attempt to quantify the fecal microbiome concentration. Instead 1g/ml fecal suspension was gavaged.
In a previous investigation, we have successfully used this approach to engraft mice with human fecal microbiota from different donors [2]. The method of using a suspension of fecal material in saline without quantifying total bacteria is similar to procedures used in human FMT [3]. Moreover, human samples were homogenized and aliquoted to ensure mice received similar fecal microbiota for successive gavages.
The fecal samples were frozen till they were deployed. The authors must explain the survival and revival of the microbiome during the freeze thaw. May be my ignorance, but, it is my understanding that normal microbes will not survive the gastric environment. But the results demonstrated the microbes indeed survived the gastric environment. However, did the gastric environment alter the population composition? An explanation will add value to this study.
Human fecal samples were frozen after they were produced and subsequently stored at -80 C. Frozen samples were homogenized and aliquoted to ensure only 1 freeze/thaw cycle at gavage. The authors agree that the gastric environment may change the microbiota composition. However, this gavage approach has been shown to be successful for FMT to humanize mice [2,4]. As shown in this work, and our previous work [2], human to mouse FMT does not have complete fidelity and the gastric environment likely contributes to this.
The water contained 1g/ml Amp. Did this antibiotic regimen continue through out the 22 weeks period. It must be explicitly stated under M&M section.
Mice were treated with ampicillin only during the antibiotic depletion portion of the study. This point has been clarified in the text.
Statistical Analyses:
I am not qualified to comment on the statistical data analyses methodology. However, I am knowledgeable to understand the data. The following points need clarifications.
How the number of animals needed for each group was selected/arrived at?
For the genomic profiling, did they pool the fecal samples?
Because the effects of the FMT and diet on our primary endpoints are not known, the study was powered using oral glucose tolerance data from previous work that investigated diet on this endpoint. With N=24 for each diet/FMT group for assessment of oral glucose tolerance, we determined have >80% power to detect a difference of 3.9 relative area under the curve units (k sample means, α = 0.05, σ = 4.8). Genomic profiling of the human fecal samples was determined from the homogenized sample. These points have been clarified in the text.
Results:
The results and the subsequent discussion sections are clearly presented.
Line 316: Body weight gain:
Is it body weight or body weight gain (rate). The animals lost weight during the first two weeks of the 22 wk tx period (figure 2A). But what happened during the native microbiome depletion period? Did the animals loose weight? So, this body weight loss is from the start of FMT or the start of the Ab treatment?
The mice lost weight during the antibiotic regimen (first two weeks), but gained weight throughout reminder of the study.
Is 12 hours post antibiotic withdrawal provide enough time to get the body rid off the antibiotics? How did this treatment affect the fecal microbiome establishment? I believe gastric environment combined with this short (12h) gap may be too much of an assault on the fecal microbiome population.
The reviewer is correct. In a previous study, we found that a single gavage after 12 hours after completion of the antibiotic regimen was not sufficient for successful engraftment [2]. Therefore, the mice were gavaged three more times every seven days.
Conclusion: Line 856: Although are hypothesis- May be a typo?
This line states 'hypothesis was incorrect'. I believe the authors meant to state the experimental data disproved our hypothesis or did not support our hypothesis. Just a thought.
We have incorporated the reviewer’s suggestion into the text.
Reviewer 2 Report
Basal diet determined long-term composition of the gut microbiome and mouse phenotype to a greater extent than fecal microbiome transfer from lean or obese human donors
By Daphne M Rodriguez et al (Corresponding authors: Abby D Benninghoff, Korry J Hintze)
Submitted to Nutrients (Editorial No: nutrients-538485)
General Comments
This manuscript represents a bold attempt to disentangle interrelationships of 1. basal diet, 2. gut microbiome composition, and 3. associated disease (obesity, hypertension, coronary heart disease, type 2 diabetes (T2D) and colorectal and other cancers). The main emphasis of the work is on obesity, and a particular purpose is to get beyond phenotypic associations and to identify causal relationships.
The influence of gut microbiota obtained from obese and lean human donors following transplantation into mice was explored under the mice receiving 3 different diets: AIN, optimized for rodents; DIO, diet inducing obesity; and TWD, total Western diet modelling US nutrient intake. The experimental diets were strictly defined. This approach was chosen in order to determine, to what extent obesity depends on residual or changed gut microbiome, or on dietary components. Mice received fecal microbial transfer (FMT) from lean and obese human donors after a 1-week residual microbiota depletion by broad spectrum antibiotic cocktails. Besides various body measurements of the animals (weight, body-weight index, fat distribution, glucose tolerance test) their gut microbiomes were prospectively determined (by 16S rRNA sequencing) and compared after classification into operational taxonomic units (OTUs). Various bio-statistical methods were used to support the degree of significance of the data obtained.
It was found that food and energy intake and body weight gain depended on diet more than the type of FMT received. Similarly, the glucose tolerance of mice was mainly affected by the diet. Mice receiving human FMTs underwent considerable changes of their microbiome during the experiment, again mainly dependent on the diet received. Limitations of the study, such as diversity of the gut microbiome of humans of different phenotype (lean vs obese) or a still apparent lack of discovering functional causality of the microbiomes analysed (by meta-genomic techniques) for human pathogenesis are clearly recognized and spelled out.
Specific Comments
Line
2 Reconsider Title, e.g. ‘Basal diet as a determinant of gut microbiome composition and phenotype of mice following fecal microbiome transfer from lean or obese human donors’, or similar.
18 Consider reading: … can alter the gut microbiome and be associated with…
22 … high fat diet inducing obesity (DIO)…
25 … Prior to FMT, the residual (normal) gut microbiome was depleted …
26 … human donor type microbiomes did not significantly affect…
28 to 29. The sentence is convoluted and should be rephrased.
35 … Obesity rates in humans have increased…
54 … short chain fatty acids… The importance of SCFAs for disease phenotypes in humans may be expanded upon slightly. Consider citation of (besides many others):
Differding M, Benjamin-Neelon S, Ostbye T, Hoyo C, Mueller N. Association of Early Introduction to Solid Foods with Infant Gut Microbiota Abundance, Overweight/obesity, and Short Chain Fatty Acids at Ages 3 and 12 Months (OR01-06-19). Curr Dev Nutr. 2019 Jun 13;3(Suppl 1). pii: nzz041.OR01-06-19.
Liu HY, Walden TB, Cai D, Ahl D, Bertilsson S, Phillipson M, Nyman M, Holm L. Dietary Fiber in Bilberry Ameliorates Pre-Obesity Events in Rats by Regulating Lipid Depot, Cecal Short-Chain Fatty Acid Formation and Microbiota Composition. Nutrients. 2019 Jun 15;11(6). pii: E1350.
Sanna S, van Zuydam NR, Mahajan A, Kurilshikov A, Vich Vila A, Võsa U, Mujagic Z, Masclee AAM, Jonkers DMAE, Oosting M, Joosten LAB, Netea MG, Franke L, Zhernakova A, Fu J, Wijmenga C, McCarthy MI. Causal relationships among the gut microbiome, short-chain fatty acids and metabolic diseases. Nat Genet. 2019 Apr;51(4):600-605.
60 … not others [12]. Recent meta-analyses…
91 … have a bigger influence… than other factors…
129 … are summarized in Table S1.
178 The treatment of mice with broad-spectrum antibiotic cocktails over a week (lines 154-166) is quite rigorous and apparently had major side effects on the mice. Some more details should be provided how quickly the ‘remaining’ mice recovered and also whether their gut content was sterile after the microbiota depletion.
213 Please read: … in Tris-HCl –EDTA buffer, pH… (indicate).
216 Table S3. This could be condensed by omitting the columns ‘Ion Torrent Primer A-Key’, ‘Key’, ‘GAT’, and ‘Primer sequence (5’-3’)’ and expanding the legend accordingly. Alternatively, a Suppl Fig. could be considered, diagrammatically showing the individual components in context.
223 … TE buffer… Define at first mentioning.
238 Clarify the use of ‘uchime61’ and provide a reference.
242 Please rephrase for clarification.
264 The log10 transformations were also used for clarity of Figures.
304 to 661. In general, the data presentation and assessment are convincing.
309f Fig. S1. P values in text and figure differ. Please check and correct as appropriate. [This comment also relates to other Figures containing P values, e.g. Figure 2-4, 10, S2.]
372 The calculation of the glucose tolerance from ‘area-under-the-curve (AUC)’ should be mentioned in Methods.
410 Join text with that of line 418.
506 There is no Fig. S15. Please check and clarify.
509 Apparently, some bacterial species/families differed in mice depending on whether they received FMTs from lean or obese donors. It should be considered to transfer such discriminating bacteria separately to mice and observe the effects.
552 It should be considered to make Fig. S12 as figure in the main Text.
602 to line 603. Close gap.
663ff The Discussion is justified in concluding that in mice receiving FMTs from lean or obese human donors the diet they receive is more important in determining the metabolic phenotype they develop than the composition of the FMT received or the microbiome they develop, and that the diet can overtake donor effects.
694 … a large community of S24-7 bacteria, which …
713ff See comment line 509. In order to obtain more clarity in this area of microbiome research, mice could be reconstituted with ‘simplified intestinal microbiota’ (SIM) which would allow the study of how diet affects microbe-diet interaction, and possibly the host’s metabolic phenotype. [Kovatcheva-Datchary P, Shoaie S, Lee S, Wahlström A, Nookaew I, Hallen A, Perkins R, Nielsen J, Bäckhed F. Simplified Intestinal Microbiota to Study Microbe-Diet-Host Interactions in a Mouse Model. Cell Rep. 2019 Mar 26;26(13):3772-3783.e6.]
773 … either a chow diet of a high-fat, high-sugar diet…
841 This limitation is well spotted. It could be elucidated further by selective feeding of SIMs (see comment line 713) with known metabolism which may contribute to pathogenic host metabolism.
856 Please check the text and rephrase.
865 The data of the authors are supported by findings of refs 18, 27, 66 and their own previous publications [28, 44, 65].
1017 … obesity…
Author Response
Response to Reviewer 2
General Comments
This manuscript represents a bold attempt to disentangle interrelationships of 1. basal diet, 2. gut microbiome composition, and 3. associated disease (obesity, hypertension, coronary heart disease, type 2 diabetes (T2D) and colorectal and other cancers). The main emphasis of the work is on obesity, and a particular purpose is to get beyond phenotypic associations and to identify causal relationships.
The influence of gut microbiota obtained from obese and lean human donors following transplantation into mice was explored under the mice receiving 3 different diets: AIN, optimized for rodents; DIO, diet inducing obesity; and TWD, total Western diet modelling US nutrient intake. The experimental diets were strictly defined. This approach was chosen in order to determine, to what extent obesity depends on residual or changed gut microbiome, or on dietary components. Mice received fecal microbial transfer (FMT) from lean and obese human donors after a 1-week residual microbiota depletion by broad spectrum antibiotic cocktails. Besides various body measurements of the animals (weight, body-weight index, fat distribution, glucose tolerance test) their gut microbiomes were prospectively determined (by 16S rRNA sequencing) and compared after classification into operational taxonomic units (OTUs). Various bio-statistical methods were used to support the degree of significance of the data obtained.
It was found that food and energy intake and body weight gain depended on diet more than the type of FMT received. Similarly, the glucose tolerance of mice was mainly affected by the diet. Mice receiving human FMTs underwent considerable changes of their microbiome during the experiment, again mainly dependent on the diet received. Limitations of the study, such as diversity of the gut microbiome of humans of different phenotype (lean vs obese) or a still apparent lack of discovering functional causality of the microbiomes analyzed (by meta-genomic techniques) for human pathogenesis are clearly recognized and spelled out.
Specific Comments
Line
2 Reconsider Title, e.g. ‘Basal diet as a determinant of gut microbiome composition and phenotype of mice following fecal microbiome transfer from lean or obese human donors’, or similar.
We appreciate the suggestion. However, we prefer to keep the original title as it summarizes the main conclusion of the paper as a declarative statement.
18 Consider reading: … can alter the gut microbiome and be associated with…
This sentence has been edited for clarity.
22 … high fat diet inducing obesity (DIO)…
This sentence has been edited for clarity.
25 … Prior to FMT, the residual (normal) gut microbiome was depleted …
This sentence has been edited for clarity.
26 … human donor type microbiomes did not significantly affect…
This sentence has been edited for clarity.
28 to 29. The sentence is convoluted and should be rephrased.
This sentence has been edited for clarity.
35 … Obesity rates in humans have increased…
The text has been changed per the reviewer suggestion.
54 … short chain fatty acids… The importance of SCFAs for disease phenotypes in humans may be expanded upon slightly. Consider citation of (besides many others):
Differding M, Benjamin-Neelon S, Ostbye T, Hoyo C, Mueller N. Association of Early Introduction to Solid Foods with Infant Gut Microbiota Abundance, Overweight/obesity, and Short Chain Fatty Acids at Ages 3 and 12 Months (OR01-06-19). Curr Dev Nutr. 2019 Jun 13;3(Suppl 1). pii: nzz041.OR01-06-19.
Liu HY, Walden TB, Cai D, Ahl D, Bertilsson S, Phillipson M, Nyman M, Holm L.Dietary Fiber in Bilberry Ameliorates Pre-Obesity Events in Rats by Regulating Lipid Depot, Cecal Short-Chain Fatty Acid Formation and Microbiota Composition. Nutrients. 2019 Jun 15;11(6). pii: E1350.
Sanna S, van Zuydam NR, Mahajan A, Kurilshikov A, Vich Vila A, Võsa U, Mujagic Z, Masclee AAM, Jonkers DMAE, Oosting M, Joosten LAB, Netea MG, Franke L, Zhernakova A, Fu J, Wijmenga C, McCarthy MI. Causal relationships among the gut microbiome, short-chain fatty acids and metabolic diseases. Nat Genet. 2019 Apr;51(4):600-605.
These references have been added per the reviewer suggestions
60 … not others [12]. Recent meta-analyses…
This sentence has been edited for clarity.
91 … have a bigger influence… than other factors…
The text has been changed per the reviewer suggestion.
129 … are summarized in Table S1.
The text has been changed per the reviewer suggestion.
178 The treatment of mice with broad-spectrum antibiotic cocktails over a week (lines 154-166) is quite rigorous and apparently had major side effects on the mice. Some more details should be provided how quickly the ‘remaining’ mice recovered and also whether their gut content was sterile after the microbiota depletion.
The issue of recovery is addressed in the body weight gain section in the Results. Mice did not gain weight during the depletion protocol but recovered as indicated by weight gain after the antibiotic treatment ended. The antibiotic depletion protocol used has been validated by another group (PMID: 21445311) and used successfully by our group to engraft human microbiota into mice [2].
213 Please read: … in Tris-HCl –EDTA buffer, pH… (indicate).
This issue has been addressed.
216 Table S3. This could be condensed by omitting the columns ‘Ion Torrent Primer A-Key’, ‘Key’, ‘GAT’, and ‘Primer sequence (5’-3’)’ and expanding the legend accordingly. Alternatively, a Suppl Fig. could be considered, diagrammatically showing the individual components in context.
Per reviewer instruction the table has been condensed.
223 … TE buffer… Define at first mentioning.
This issue has been addressed.
238 Clarify the use of ‘uchime61’ and provide a reference.
This issue has been addressed
242 Please rephrase for clarification.
The sentence has been rephrased for clarity.
264 The log10 transformations were also used for clarity of Figures.
This statement has been added to the text.
304 to 661. In general, the data presentation and assessment are convincing.
309f Fig. S1. P values in text and figure differ. Please check and correct as appropriate. [This comment also relates to other Figures containing P values, e.g. Figure 2-4, 10, S2.]
We recognize that the presentation of such complicated statistical analyses is challenging, and we have endeavored to make the statistical results as clear as possible within the paper. Presentation of p values for each figure are explained in the figure legends. Note, the tables below each figure (where shown) indicate overall main effects for each experimental factor. In figures, symbols associated with brackets indicate main effects for each particular diet treatment. Post hoc test results within each diet group comparing FMT from lean or obese donors are presented within the figure by symbols above the box-and-whisker bars ( * p<0.05, ** p<0.01, *** p<0.001, **** p<0.0001, ns=not significant).
Within the text, we aimed to clarify when we were presenting main effect results (e.g., any p-values for overall diet effects) or for post-hoc comparisons within a diet between FMT from lean or obese donors. For clarity, we have added specifications for each type post-hoc test, either diet or FMT, as outlined in lines 274-277.
Additionally, we have specified p-values obtained for beta-diversity analyses as permanova p-values,
372 The calculation of the glucose tolerance from ‘area-under-the-curve (AUC)’ should be mentioned in Methods.
We added explanatory text to the methods, see lines 186-188.
410 Join text with that of line 418.
This issue has been fixed
506 There is no Fig. S15. Please check and clarify.
This issue has been fixed
509 Apparently, some bacterial species/families differed in mice depending on whether they received FMTs from lean or obese donors. It should be considered to transfer such discriminating bacteria separately to mice and observe the effects.
The reviewer raises an important point. To address this comment, and a similar comment below, we have added additional text to the discussion section discussing how future work could investigate the contribution of specific taxa using a defined population in FMT. See lines 734-737
552 It should be considered to make Fig. S12 as figure in the main Text.
Given that there is no limitation on the number of figures in the paper, we agree that this figure can be moved into the main paper.
602 to line 603. Close gap.
This issue has been fixed
663ff The Discussion is justified in concluding that in mice receiving FMTs from lean or obese human donors the diet they receive is more important in determining the metabolic phenotype they develop than the composition of the FMT received or the microbiome they develop, and that the diet can overtake donor effects.
694 … a large community of S24-7 bacteria, which …
This issue has been fixed
713ff See comment line 509. In order to obtain more clarity in this area of microbiome research, mice could be reconstituted with ‘simplified intestinal microbiota’ (SIM) which would allow the study of how diet affects microbe-diet interaction, and possibly the host’s metabolic phenotype. [Kovatcheva-DatcharyP, Shoaie S, Lee S, Wahlström A, Nookaew I, Hallen A, Perkins R, Nielsen J, Bäckhed F. Simplified Intestinal Microbiota to Study Microbe-Diet-Host Interactions in a Mouse Model. Cell Rep. 2019 Mar 26;26(13):3772-3783.e6.]
The reviewers appreciate this suggestion and have added text to address this issue at the end of the paragraph and have cited the provided reference. See lines 734-737
773 … either a chow diet of a high-fat, high-sugar diet…
This issue has been fixed
841 This limitation is well spotted. It could be elucidated further by selective feeding of SIMs (see comment line 713) with known metabolism which may contribute to pathogenic host metabolism.
Please see previous comment (line 713)
856 Please check the text and rephrase.
This issue has been fixed
865 The data of the authors are supported by findings of refs 18, 27, 66 and their own previous publications [28, 44, 65].
This text has been added to the final paragraph.
1017 … obesity…
This issue has been fixed
References for this response to reviewers
1. Rodriguez Jimenez, D.M.; Benninghoff, A.D.; Hintze, K.J. Impact of basal diet on obeseity phenotype of recipient mice following fecal microbiome transfer from obese or lean human donors: Appendix D. Microbiome sequencing data. Utah State University Digital Commons: Logan, Utah, 2018; doi.org/10.15142/T33W7W.
2. Hintze, K.J.; Cox, J.E.; Rompato, G.; Benninghoff, A.D.; Ward, R.E.; Broadbent, J.; Lefevre, M. Broad scope method for creating humanized animal models for animal health and disease research through antibiotic treatment and human fecal transfer. Gut Microbes 2014, 5, 183-191, doi:10.4161/gmic.28403.
3. Vrieze, A.; Van Nood, E.; Holleman, F.; Salojarvi, J.; Kootte, R.S.; Bartelsman, J.F.; Dallinga-Thie, G.M.; Ackermans, M.T.; Serlie, M.J.; Oozeer, R., et al. Transfer of intestinal microbiota from lean donors increases insulin sensitivity in individuals with metabolic syndrome. Gastroenterology 2012, 143, 913-916 e917, doi:10.1053/j.gastro.2012.06.031.
4. Turnbaugh, P.J.; Ridaura, V.K.; Faith, J.J.; Rey, F.E.; Knight, R.; Gordon, J.I. The effect of diet on the human gut microbiome: a metagenomic analysis in humanized gnotobiotic mice. Sci Transl Med 2009, 1, 6ra14, doi:10.1126/scitranslmed.3000322.